# Prevalence and factors associated with human *Taenia solium* taeniosis and cysticercosis in twelve remote villages of Ranomafana rainforest, Madagascar

**Anjanirina Rahantamalala**[1]*, **Rado Lalaina Rakotoarison**[1], **Emma Rakotomalala**[1‡], **Mahenintsoa Rakotondrazaka**[1], **Jaydon Kiernan**[2‡], **Paul M. Castle**[2], **Lee Hakami**[2], **Koeun Choi**[2‡], **Armand Solofoniaina Rafalimanantsoa**[3,4], **Aina Harimanana**[5], **Patricia Wright**[6,7], **Simon Grandjean Lapierre**[8¤a¤b], **Matthieu Schoenhals**[1], **Peter M. Small**[6,8], **Luis A. Marcos**[8,9], **Inès Vigan-Womas**[1¤c]

1 Institut Pasteur de Madagascar, Immunology of Infectious Diseases Unit, Antananarivo, Madagascar, 2 Renaissance School of Medicine, Stony Brook University, Stony Brook, New York, United States of America, 3 Institut Pasteur de Madagascar, Helminthiasis Unit, Antananarivo, Madagascar, 4 Ministry of Public Health, Antananarivo, Madagascar, 5 Institut Pasteur de Madagascar, Epidemiology and Clinical Research Unit, Antananarivo, Madagascar, 6 Centre ValBio, Ranomafana, Ifanadiana, Madagascar, 7 Department of Anthropology, Stony Brook University, Stony Brook, New York, United States of America, 8 Global Health Institute, Stony Brook University, Stony Brook, New York, United States of America, 9 Division of Infectious Diseases, Department of Medicine, Department of Microbiology and Molecular Genetics, Stony Brook University, Stony Brook, New York, United States of America

¤a Current address: Immunopathology Axis, research Center, University Hospital Centre, Montreal, Canada
¤b Current address: Microbiology, Infectious Diseases and Immunology Department, University of Montreal, Canada
¤c Current address: Institut Pasteur de Dakar, Immunophysiopathology and Infectious Diseases Department, Dakar, Senegal
‡ Unavailable
* anjanirina@pasteur.mg

**Data Availability Statement:** 1-All the files of the ten T. solium Asian and African/American

## Abstract

### Background

Infections with the tapeworm *Taenia solium* (taeniosis and cysticercosis) are Neglected Tropical Diseases (NTD) highly endemic in Madagascar. These infections are however underdiagnosed, underreported and their burden at the community level remains unknown especially in rural remote settings. This study aims at assessing the prevalence of *T. solium* infections and associated risk factors in twelve remote villages surrounding Ranomafana National Park (RNP), Ifanadiana District, Madagascar.

### Methodology

A community based cross-sectional survey was conducted in June 2016. Stool and serum samples were collected from participants. Tapeworm carriers were identified by stool examination. *Taenia* species and *T. solium* genotypes were characterised by PCR and sequencing of the mitochondrial *cytochrome c oxidase* subunit 1 (*cox1*) gene. Detection of specific

genotypes identified during this study are available from the GenBank database with the following identifiers and accession number (AC):
IFAI_Mangevo_051, AC: MT947371
IFAI_Kianjanomby_332, AC: MT947372
IFAI_Kianjanomby_365, AC: MT947373
IFAI_Ampitambe_378, AC: MT947374
IFAI_Ankazotsara_452, AC: MT947375
IFAI_Torotosy_564, AC: MT947376
IFAI_Torotosy_573, AC: MT947377
IFAI_Ampitavanana_641, AC: MT947378
IFAI_Sahavoemba_182, AC: MT947379
IFAI_Sahavoemba_190, AC: MT947380 2-All relevant data are within the manuscript and its Supporting Information files 3-Some additional data cannot be shared publicly because of confidentiality; it studied specific area and targeted generally all the people leaving in each village. All villages studied are very remote and very isolated whose population does not move and can be easily identified even through anonymous identifiers. Data are available from the Institut Pasteur de Madagascar Data Access / Ethics Committee (contact: ipm@pasteur.mg) for researchers who meet the criteria for access to confidential data.

**Funding:** This study received financial support from the Global Health Institute of Stony Brook University (https://www.stonybrook.edu/ghi) the David E. Rogers Student Fellowship Award (New York Academy of Medicine, https://www.nyam.org/fellows-grants) USA, the ValBio Centre Ranomafana Madagascar (https://www.stonybrook.edu/commcms/centre-valbio), and the Institut Pasteur de Madagascar (http://www.pasteur.mg). The study's funders had a role in study design, data collection, data analysis, data interpretation, or writing of the manuscript.

**Competing interests:** The authors have declared that no competing interests exist. Authors Emma Rakotomalala, Jaydon Kiernan, and Koeun Choi were unavailable to confirm their authorship contributions. On their behalf, the corresponding author has reported their contributions to the best of their knowledge.

anti-cysticercal antibodies (IgG) or circulating cysticercal antigens was performed by ELISA or EITB/Western blot assays.

## Principal findings

*Of the 459 participants* with paired stool and blood samples included ten participants from seven distinct villages harbored *Taenia* spp. eggs in their stools samples DNA sequencing of the *cox1* gene revealed a majority of *T. solium* Asian genotype (9/10) carriage. The overall seroprevalences of anti-cysticercal IgGs detected by ELISA and EITB were quite similar (27.5% and 29.8% respectively). A prevalence rate of 12.4% of circulating cysticercal antigens was observed reflecting cysticercosis with viable cysts. Open defecation (Odds Ratio, OR = 1.5, 95% CI: 1.0–2.3) and promiscuity with households of more than 4 people (OR = 1.9, 95% CI: 1.1–3.1) seem to be the main risk factors associated with anticysticercal antibodies detection. Being over 15 years of age would be a risk factor associated with an active cysticercosis (OR = 1.6, 95% CI: 1.0–2.7). Females (OR = 0.5, 95% CI: 0.3–0.9) and use of river as house water source (OR = 0.3, 95% CI: 0.1–1.5) were less likely to have cysticercosis with viable cysts.

## Conclusions/Significance

This study indicates a high exposure of the investigated population to *T. solium* infections with a high prevalence of cysticercosis with viable cysts. These data can be useful to strengthen public health interventions in these remote settings.

### Author summary

*Taenia solium* infections in humans (taeniosis and neurocysticercosis) and in pigs (cysticercosis) are endemic in Madagascar presenting a significant public health burden. Neurocysticercosis with localization of the parasite in the Central Nervous System is the most severe and frequent form of parasitic brain diseases in humans and responsible of thousands of worldwide deaths per year. Madagascar is a *T. solium* endemic country where poor sanitation, free roaming pigs and outdoor defecation are common, and maintain the parasite transmission cycle. Little information is available regarding taeniosis/cysticercosis epidemiology in Madagascar. We carried out a community-based study to investigate the prevalence of human taeniosis/cysticercosis and associated risk factors in 12 rural remote villages of Ranomafana and Kelilalina townships (Ifanadiana district, Madagascar). Our results reveal that in 7/12 villages investigated, a high number of participants had taeniosis. Moreover, a high number of active cysticercosis cases were detected. Open defecation and promiscuity were seemed to be the main risk factors associated to *T. solium* infections. The results of this study will be useful to guide interventions in these remote settings surrounding the Ranomafana National Park.

## Introduction

*Taenia solium* taeniosis/cysticercosis are neglected parasitic tropical diseases mainly affecting people living in Central and South America, South-East Asia, Indian Subcontinent, and Sub-

Saharan Africa [1]. Also referred to as poverty-related diseases, these infections are associated with poor sanitary and hygiene conditions, open defecation, free-range pig-husbandry and lack of meat inspection [2]. Despite having been declared potentially eradicable, *T. solium* infections remain a serious health and economic problem affecting around 50 million people worldwide [3]. *T. solium* was identified as a leading cause of death from food-borne diseases, accounting for 2.8 million disability-adjusted life-years (DALYs) annually [4].

*T. solium* taeniosis and cysticercosis are zoonosis with a biological cycle maintained in the environment between humans and pigs [5]. In the natural life cycle of the parasite, humans are the only definitive hosts developing the intestinal adult form (causing taeniosis). Cysticercosis in pigs and in human (respectively the natural and the accidental intermediary host) is caused by the metacestode larval stage of *T. solium* (called cysticerci). Humans can therefore develop cysticercosis in muscles, eyes or brain. Neurocysticercosis (NCC) is the most severe presentation of this pathology and the most important parasitic disease affecting the central nervous system [6–9]. NCC may count for almost one-third of seizure disorders with an estimated 2 million people affected [10,11].

A few community-based studies were carried out in Africa to estimate the prevalence of *T. solium* infections using different diagnostic tools [12–14]. Human cysticercosis prevalence varied greatly across African countries, 0.68% to 21.63% of analyzed samples were found positive for circulating antigens while the seropositivity associated with *T. solium* antibodies ranged from 7.6% to 34.5%. The prevalence of taeniosis reported in Africa was between 0.37 and 13.8% [12]. Nevertheless, there are still little data available regarding the prevalence of *T. solium* infections and their associated risk factors for the African Continent especially in rural areas [12,15–17].

In Madagascar, the proportion of the population living below the international poverty line was estimated at 75% in 2019 [18]. Around 35% of the population do not have access to basic toilets [19] and 52% of the population (only 35% in rural areas) have access to drinking water [20]. Agriculture, mainly mixed with livestock, is still very dominant with more than seven out of ten employed individuals. In rural areas, about two-thirds of households are traditional farmers where pig breeding (mainly raised in free roaming) holds the third place [18]. Given the persistence of factors determining the spread of *T. solium* infections such as traditional pig farming, open defecation and poor meat hygiene requirements, Madagascar is among the most infested countries in the world.

Madagascar is a vast island divided into 114 Districts where taeniosis and cysticercosis epidemiological data are scarce [21]. However, the few data available underline the significant burden of *T. solium* infections in the Malagasy population. At the national level, the only and most recent sources of data on taeniosis were provided by mapping surveys (Based on Kato-Katz tests) performed through integrated national program of the Ministry of Public Health targeting some Neglected Tropical Diseases (i.e. lymphatic filariasis, schistosomiasis and soil-transmitted helminthiasis). These surveys conducted between 2011 and 2015 identified 54/114 co-endemic Districts for schistosomiasis and taeniosis of which 7 districts were endemic only to *T. solium* [22]. In 2015, an overall prevalence of 0.7% (ranging from 0.3% to 7.3%) was reported affecting 14 of the 24 Districts surveyed. Regarding cysticercosis, very little data are available in the literature and the data collected by the Ministry of Health are mainly based on clinical suspicions rarely confirmed with neuro-imaging or serological diagnosis (detection of *T. solium* cysticercal antigens by ELISA and/or anti-cysticercal antibodies by ELISA and/or Western blot/ElectroImmunoTransferBlot-EITB). Indeed, since the existence of human cysticercosis in Madagascar which was first documented by Andrianjafy [23], subsequent data have shown that approx. 20–25% of late-onset epilepsy were related to cysticercosis [24,25]. The seroprevalence of the human cysticercosis in the island ranged from 7% in Coastal regions

(Mahajanga and Toamasina) to 14–21% in Central Highlands (Ambositra, Ihosy, Mahasolo and Antsirabe) following the geographical importance of pig farming and pig cysticercosis [26–31]. These studies are unlikely to reflect the true situation as they did not cover the whole country and the remote rural areas.

The Ifanadiana District (13 townships) is one of the rural Districts of Madagascar heavily burdened by poverty and reported to have a high rate of intestinal parasites [32–34]. In 2015 however, the national mapping program reported 0% of taeniosis prevalence in 14 fokontany (the smallest administrative subdivision of Madagascar) including 21 rural villages (comprising one of the villages targeted by this study) [35]. The main objective of the current study was to determine by community based cross-sectional survey the prevalence of human *Taenia solium* taeniosis/cysticercosisis and the risk factors associated in 12 remote villages of Ranomafana and Kelilalina townships (Ifanadiana District, Madagascar).

## Methods

### Ethics statement

The study was reviewed and approved by the Ethics Committee for Biomedical Research at the Ministry of Public Health of Madagascar (N˚ 142-MSANP/CE), and by the Institutional Review Board (IRB) of the Stony Brook University (IRB#874952). The field study was conducted under the supervision of the medical team of ValBio Centre and followed ethical principles according to the Helsinki Declaration. All participants received an explanation of the study goals and procedures prior to enrolment. Written informed consent was obtained from adult participants or parent/legal guardians for the children. Participants detected positive for taeniosis were treated by local medical officers with praziquantel at 10 mg/kg in one dose according to the national guidelines. Before this treatment, in order to recover the whole worm properly and avoid dissemination of eggs in the environment, all carriers of *Taenia* were cured with niclosamide (2 g orally single dose) and with oral purgative (lactulose, 2 tablespoons) one day before and 2 hours after niclosamide. During the 3 days following the niclosamide treatment, stool samples, collected in large plastic bags, were obtained from all *Teania* carriers.

### Study design and population

This is a cross-sectional prevalence study. Overall, 164 households were randomly selected using an electronic random number generator on the village census. Household members older than 5 years and living in the study area (> 50% of their time) during the last 3 months before the beginning of the study were invited to participate. The study was conducted in the District of Ifanadiana (Vatovavy-Fitovinany Region) Southeastern Madagascar. The investigation was carried out in two rural townships: Ranomafana (21˚15'36" S and 47˚27'17" E, elevation 613m) and Kelilalina (21˚17'8" S and 47˚33'17" E, elevation 613m). The targeted villages were situated close to the ValBio Centre (CVB), a Stony Brook University Research Centre in the Ranomafana National Park. The study population was enrolled in five Fokontany, namely Vohimarina, Tsaramandroso, Mandrivany, Ampitambe and Kianjanomby. In these fokontany, twelve rural and remote villages (Ambinanindranofotaka, Mangevo, Marojano, Sahavanana, Sahavoemba, Kianjanomby, Mandrivany, Ankazotsara, Ampitambe, Bevoahazo, Torotosy, Ampitavanana) were targeted (Fig 1). These villages were chosen as they constitute the rural villages receiving biannual medical consultations by the medical team of ValBio Centre with a deworming of the entire population.

The villages included in each fokontany and the geographic coordinates (obtained using a handheld Garmin global positioning system, GPS) of the twelve villages investigated are listed in supporting S1 Table. The geo-reference data collected were used to map the study sites

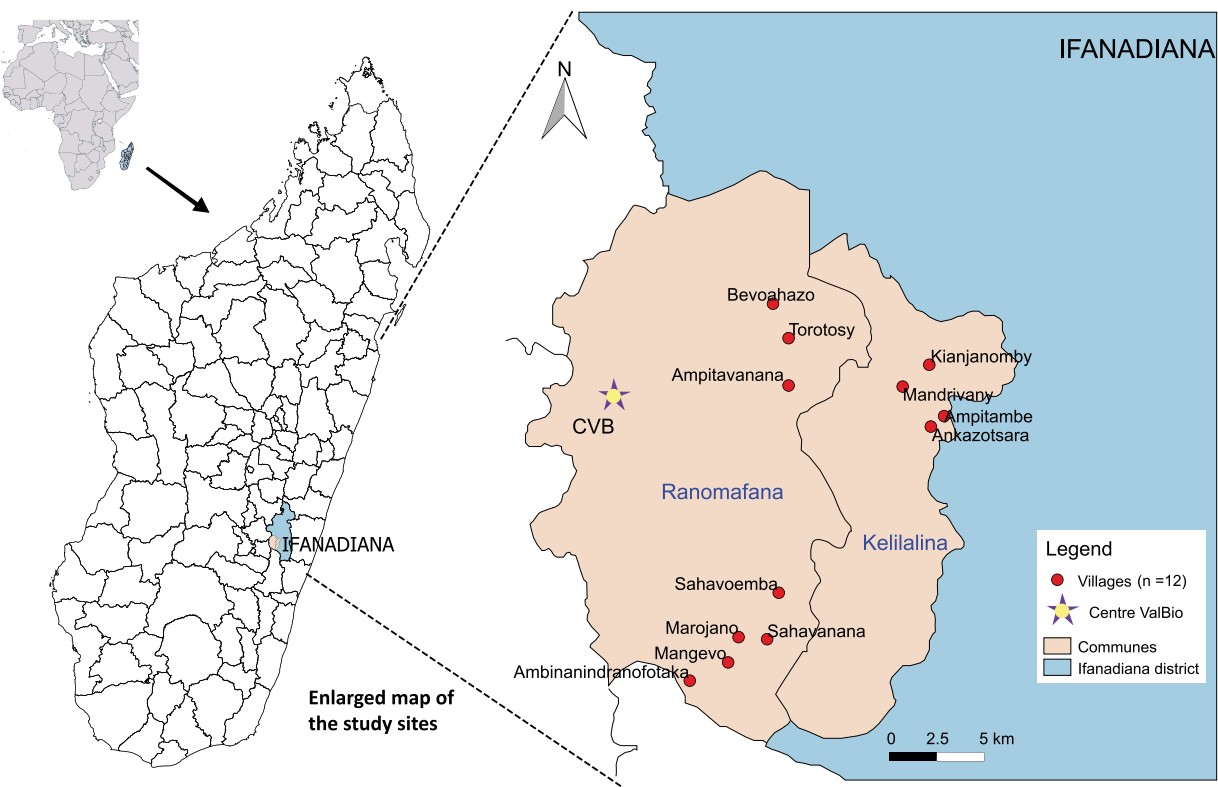

**Fig 1. Map of the study sites.** Map of Ranomafana and Kelilalina townships (Ifanadiana District) showing the twelve villages where sampling was carried out. The figure was created using the humanitarian Data Exchange (HDX) accessible from OCHA (United Nations Office for the Coordination of Humanitarian Affairs) services and using QGIS 2.8.1. https://data.humdata.org/dataset/madagascar-administrative-level-0-4-boundaries. Details on the corresponding license can be accessed via the link below: https://data.humdata.org/about/license.

using QGIS 2.14.19-Essen software (http://www.qgis.org). All twelve villages, situated around Ranomafana Rainforest Park, are only accessible after 2–7 hours of walking with no roads or paved paths. Investigated villages were found to be traditional, rural, and heavily reliant on subsistence agriculture. Village sanitation was poor, with lack of water infrastructure, common practice of open defecation and free-range pig-husbandry.

### Socio-demographic, behavioural and knowledge data collection

A community-based survey was conducted over a two-month period (June to August 2016). Results on the prevalence of soil-transmitted helminthiasis and risk factors associated were presented in another manuscript and are not further discussed here [34].

Socio-demographic features, hygiene practices and behavioural data (water source, use of soap, hand-washing, pork-eating, defecation place), knowledge about worms, some clinical manifestations suggestive of NCC (headache and seizure) and antiparasitic drug-taking were obtained using an individual questionnaire in Malagasy. Heads of households were also interviewed about the size of the household, water and sanitation conditions and pig breeding. All participants were informed on the risks associated with *T. solium* infections.

### Stool sampling and *Taenia* spp. detection by microscopy

After signing the consent form, stool samples were collected from each participant in a sealed 50 mL cup for parasitological analysis. Approximately 100–250 mg of stools was conserved at

room temperature in a 2.5% potassium dichromate solution (VWR chemicals, Ref. 26784.231) until PCR analysis. Fresh stool samples were analyzed for *Taenia* spp. eggs using two parasitological techniques, Kato-Katz method (KK) and the Spontaneous Sedimentation Technique (SS) [36,37]. Preliminary diagnoses were performed by medical students (Stony Brook University, USA) under the supervision of a trained parasitologist from Stony Brook University and were confirmed by trained parasitologists from Institut Pasteur de Madagascar.

### *Taenia* species identification

*Taenia* species identification (*T. solium*, *T. saginata* and *T. asiatica)* from stool samples of tapeworm carriers was performed by sequencing the mitochondrial cytochrome C oxidase subunit 1 (*cox1*) gene following PCR amplification. Copro-DNAs were extracted from approx. 150 mg of the eleven stools positive for *Taenia* eggs conserved in potassium dichromate. Briefly, after three washes with PBS (DPBS 1X, phosphate buffer saline, Invitrogen), genomic DNA extraction was subsequently realized using the QIAamp DNA Mini Stool kit (Qiagen, Germany, Ref. 51504) according to the manufacturer's instructions. DNA was eluted in 200μl of buffer and stored at -20˚C until use.

Briefly, *cox1* gene sequences from geographically different areas and representative of *T. asiatica* (GenBank accession No. AB107235.1); *T. saginata* (GenBank accession No. AB107246.1); *T. solium* Asian genotype (GenBank accession No. AB066488); and *T. solium* African/American genotype (GenBank accession No. AB066492.1) were obtained from EMBL/Genbank databases and aligned using the Clustalw software [38]. A region of 627 base pairs (starting at the position 472 of the completed *T. solium cox1* gene, GenBank accession No. AB491986.1), totally similar within the same species and containing enough polymorphisms to differentiate between *Taenia* species and *T. solium* genotypes, was selected to design common primers (Forward: 5'-GACTAATATATTTTCTCGTAC-3' and reverse: 5- GACA TAACATAATGAAAATG-3'). The details of the polymorphic region with the primer delimitation are shown in supporting S1 Fig.

In a first step, primer pairs described by Yamasaki et *al.* [39] were used to perform *Taenia* spp. specific PCR. However, PCR amplification performed with these primers used either in multiplex or simplex assays showed slight cross-amplifications: *T. asiatica* primers amplified both *T. saginata* and *T. solium* DNAs and *T. solium* Asian genotype primers amplified *T. asiatica* DNA. Therefore, common primer pairs able to amplify the three *Taenia* species were designed, and PCR product were directly sequenced. PCR reactions using these common primers were performed in a total reaction volume of 20 μL containing 0.5 μM of each primer (Sigma Aldrich, Germany), 0.05U of Phusion HF DNA polymerase (New England Biolabs, M0530S), 1X Phusion HF Buffer including 200 μM dNTPs (New England Biolabs, B0518S) and 5 μl of genomic DNA template. PCR cycling condition was carried out according to the following program: initial denaturation step at 98˚C, 30 sec followed by 35 amplification cycles (denaturation at 98˚C, 10 sec; annealing at 58˚C, 30 sec and elongation at 72˚C, 30 sec) and a final elongation step at 72˚C, 10min. All PCR reactions were run on GeneAmp PCR System 9700 Applied Biosystem thermal cycler. Parasitic DNA extracted and purified from *T. solium* cysticerci isolated from pig was used as intra-run positive control and distilled water as negative control.

### *T. solium cox1* gene sequencing

The amplicons obtained after *cox1* gene amplification by PCR (627 bp) were sequenced in both directions by Genewiz (France). Sequence chromatograms were analyzed using BioEdit software. By removing the beginnings and ends of sequences that were illegible after

sequencing, a region of 474 base pairs (starting at the position 601 of the completed *cox1* gene sequence) for 10 out of 11 *Taenia* carriers was finally analyzed for *Taenia* species identification. The sequences were compared with the 627 base pairs reference regions of each species used beforehand to design the primers.

All nucleotide sequences obtained in the present study were made openly accessible under the following accession numbers: MT947371 (IFAI_Mangevo_051), MT947372 IFAI_Kianjanomby_332), MT947373 (IFAI_Kianjanomby_365), MT947374 (IFAI_Ampitambe_378), MT947375 (IFAI_Ankazotsara_452), MT947376 (IFAI_Torotosy_564), MT947377 (IFAI_-Torotosy_573), MT947378 (IFAI_Ampitavanana_641), MT947379 (IFAI_Sahavoemba_182), MT947380 (IFAI_Sahavoemba_190).

### *T. solium* genotypes identification and phylogenic analysis

For the *T. solium* genotypes and phylogenic analysis, the ten sequences obtained in this study were also compared with *T. solium* Asian and African/American genotype sequences available in EMBL/Genbank databases. Their accession numbers and the corresponding native countries are as followed: Madagascar: AB781355 to AB781361[40]; FM958305 to FM958317 [41]; Japan: AB516957, AB494702.1; India: KC709810.1; Thailand: AB066487; Indonesia: AB631045.1; Nepal: AB491986; AB524780.1; China: AB066486.1; Korea: DQ089663.1; Mexico: AB066490, FN995657; Cameroon: FN995666.1; Brazil: AB066492.1; Ecuador: AB066491.1; and Tanzania: AB066493. The evolutionary analyses were conducted and a tree was constructed using the Maximum Likelihood method and Jukes-Cantor model [42] using the MEGA X software [43] with 100 bootstrap replicates for reliability tests.

### Blood collection

After signing the consent form, blood sampling was performed for each participant in order to carry out serological diagnosis of *T. solium* infections according to the flow diagram represented in Fig 2. Finger prick blood samples (300–500 µL) were collected in microtubes (Sarstedt, 20.1344). The serum was obtained after 10 minutes of centrifugation at 12 000 rpm, transported at 4˚C (cool boxes) within 24 hours at the CVB where they were stored at -20˚C. At the end of the survey, all serum samples were transferred and stored to the Institut Pasteur de Madagascar (IPM) at -20˚C until use for the serological tests.

### *T. solium* glycoprotein antigen preparations

An in-house *T. solium* glycoprotein antigen suspension was prepared for anti-cysticercal antibody (IgG) detection by ELISA and EITB assays. *T. solium* cysticerci were harvested from infected pork meat obtained in slaughterhouse (Antananarivo) and stored at -80˚C in PBS buffer (DBPS 1X, Sigma Aldrich D8537) until use. Lentil lectin *T. solium* metacestode glycoproteins (LLGP) antigens were extracted and purified according to the method initially described by Tsang et *al.*, 1989 with slight modifications [44]. All extraction and purification steps have not been changed except for the foamy lipid residues above the pellet that was removed carefully with a spatula without using Freon ($CHFCl_2$) and the urea solubilization of the pellet was not carried out.

### Anti-cysticercal antibody (IgG) detection by enzyme-linked immunosorbent assay (Ab-ELISA)

In-house Ab-ELISA assay was performed as previously reported [27]. Briefly, ELISA 96-well microplates (Nunc 3455) were coated with 100 µL/well of previously prepared *T. solium*

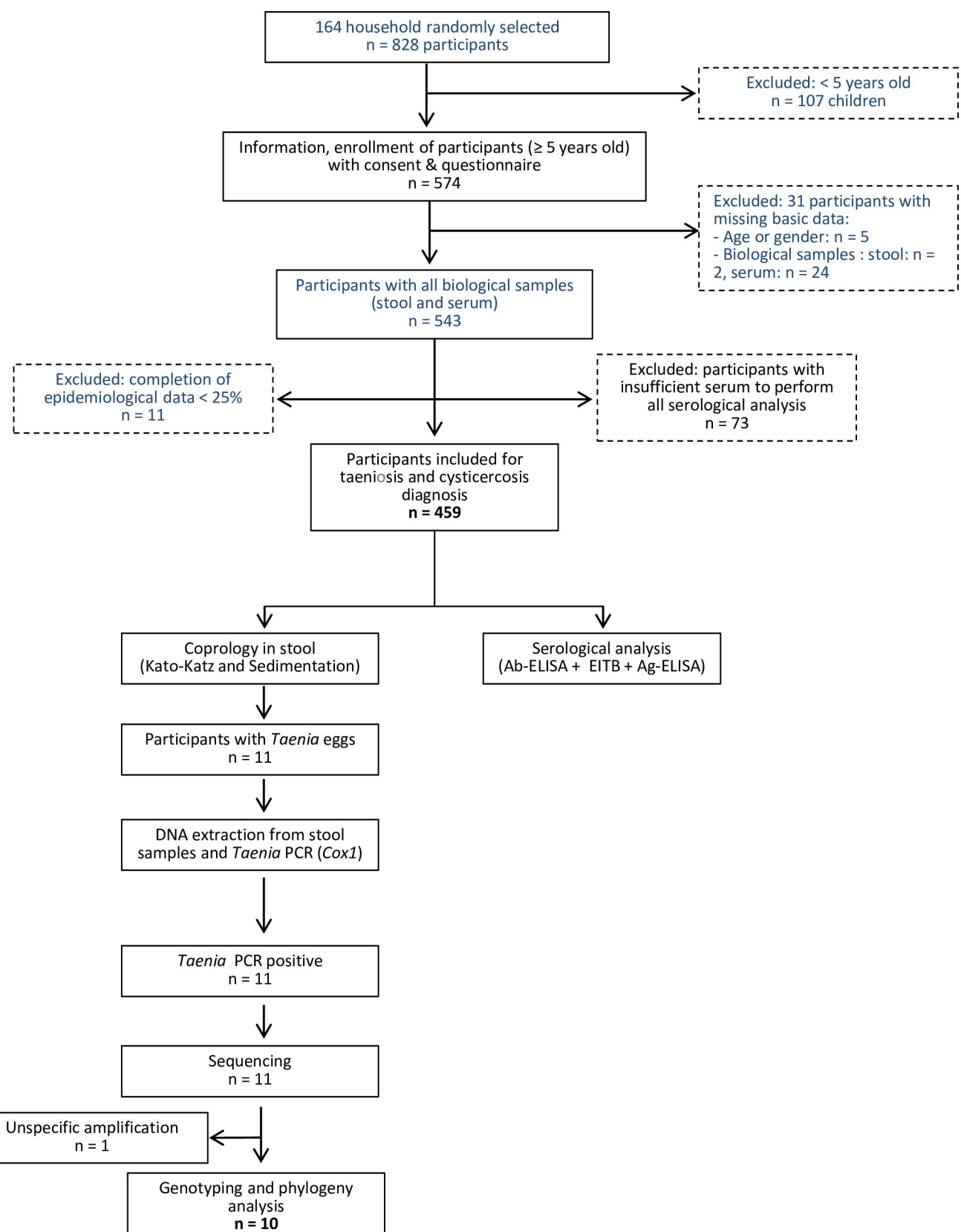

**Fig 2. Flow diagram for taeniosis (coprology and molecular analysis) and cysticercosis (serological assays) performed during this study.**

glycoprotein antigens extract at 0.1 μg. mL$^{-1}$ in DPBS 1X and incubated overnight at 4˚C. After three washing steps in washing solution (1X DPBS, 0,05% v/v Tween 20), non-specific absorption was blocked by adding 150 μL/well of blocking buffer (DPBS 1X, 0.05% Tween 20, 1% casein) and plates were incubated for 2h at 37˚C. After three additional washing steps, 100 μL of serum samples, tested in duplicate and diluted 1:200 in blocking solution, were added per wells and incubated at 37˚C for 2 h. Well characterized serum samples of patients from the neuropsychiatry department of Befelatanana Hospital (tested by EITB assay as described below and by neuroimaging) and serum samples from voluntary donors (tested by EITB assay) as positive and negative controls respectively were systematically included in each plate. Plates were then washed five times and incubated at 37˚C for 1h30 with 100 μL/well of Human antibody anti-IgG conjugated with peroxidase (Sigma Aldrich A8419) diluted 1:10 000 in blocking solution. Plates were then washed five times before a 20 min incubation at 37˚C with 100 μL/well of substrate [(4.5 mM OPD, 3.6% $H_2O_2$) in 0.2 M citric acid buffer, pH 5.5, Sigma Aldrich, Ref. P9029 and 251275]. The enzymatic reaction was stopped by addition of 100μL/well of 2.5N $H_2SO_4$ acid. The optical density (OD) was determined at 492 nm using microplate reader (Labsystem multiscan Plus). The negative cut-off value (0.3 OD) was calculated as the mean optical density (OD) of 38 sera (obtained from European healthy subjects living in *T. solium* infections non-endemic areas, France) plus 3 standard deviations. A previous study done in Madagascar has reported a specificity of 97.4% and a sensitivity of 96.3% of this in-house Ab-ELISA assay [27].

## *T. solium* cysticerci antigens detection by ELISA (Ag-ELISA)

The serum samples were tested for the presence of excretory secretory circulating antigens of the metacestode of *T. solium* using the commercially available cysticercosis Antigen-ELISA Kit (ApDia Ltd., Turnhout, Belgium) based on the B158/B60 monoclonal antibodies enzyme-linked immunosorbent assay (ELISA) [45]. The ELISA was also used and carried out according to the manufacturer's instructions and the OD was read at 450nm. The Ag-index was calculated following the recommended cut-offs: an Ag-index less than 0.8 classified as negative result, an Ag-index greater than or equal to 1.3 considered as positive and values in between classified as doubtful. The reported sensitivity and specificity of the Ag-ELISA for detecting active cysticercosis in humans were 90% (95% CI: [80–99%]) and 98% (95% CI: [97–99%]), respectively [45,46].

## Enzyme-linked Immunoelectro-Transfer Blot (EITB) assay

The EITB assay was performed as previously described by Tsang et *al.*, 1989 [44]. The specificity and sensitivity of the EITB for detecting exposure to the parasite was reported ranging from 97% to 100% and from 97% to 98% respectively [44,46]. The usual concentration for LLGP-antigens in-house EITB assay was about 30ng/μL per mm of membrane. Briefly, each strip membrane was incubated for 2h at 37˚C with human diluted (1:10) sera. The strips were thereafter incubated for 1h at 37˚C with diluted (1:6000) alkaline phosphatase-conjugated goat anti-human IgG (Sigma Aldrich, A1543). Immune reactivity was detected using BCIP/NBT blue substrate solution (Sigma Aldrich, B3804). Among the seven glycoprotein (GP) bands commonly recognized by antibodies from sera of patients with cysticercosis, only the immune reactivity against one or several GP < 50 kDa (*i.e.* GP42-39, GP24, GP21, GP18, GP14, and GP13) were considered as positive. As a precaution, tests giving a single 50kDa band were not considered as a positive result in this study. Indeed some studies have shown the lack of specificity of a single positive 50kDa band in EITB assay for cysticercosis [47–49].

## Statistical analysis

Descriptive analyses on the study population were first conducted. Prevalence of *Taenia* infections was estimated as the number of samples tested positive divided by the total number of participants provided samples. Data were analyzed with STATA 15 software. Due to the lack of the specificity of Ab-ELISA [47,50–53], only the prevalence obtained with EITB and Ag-ELISA was used for the association analysis using STATA software. Variables showing more than 25% of missing data were not included into the statistical analysis. A univariate analysis using logistic regression was performed to determine associations between the prevalence and each risk factor. A *p* value $\leq$ 0.2 was included in a model (Initial model) for multivariable logistic stepwise regression analysis. A backward deletion method was used to eliminate the factor one at a time considering a $p \leq 0.05$ as statistically significant (Final model). A Confidence Interval of 95% (95% CI) and a significance level of 0.05 were performed for the association analysis and the odds ratio calculation.

## Results

### Participant characteristics

People of all age and both genders, living in the targeted villages and willing to participate to the study were enrolled. A total of 574 willing participants ($\geq$ 5 years old) from 12 villages belonging to 164 households were included in this study. Stool and blood samples were obtained from 543/574 (92.7%) participants following the exclusion of thirty-one participants with missing basic data (age or gender) and missing biological samples (stool or serum). Seventy-three and eleven participants were further excluded for the analysis respectively due to the insufficient volume of sera to perform all three serological tests used for cysticercosis diagnosis (*i.e.*, Ab-ELISA, ETIB and Ag-ELISA tests) and due to the completion of the epidemiological data (<25%) for the statistical analysis. The final study population consisted of 459 participants surveyed having all epidemiological and serological data available. Fig 2 shows the flowchart describing the enrolment and analysis performed.

Socio-demographic characteristics and health/hygiene conditions of the study population are described in Table 1. Genders were equally represented (235 males and 224 females, sex ratio 1.05). The median age was 23.9 years, ranging from 5 to 82 years old with a high proportion of adults over 15 (56%) and just over a quarter (27.7%) were children between 5 and 10 years old. The households were composed mainly of large families (52.1% of households with 4 to 6 people, median 6.4 people) and the adult population was mostly composed of farmers (54.5%). The highest education was the elementary level (68.6%), of which 38.3% were adults. Thirteen-point five percent of adults $\geq$ 15 years old never attended school. More than two-thirds of participants lived 6 km or more from the main national road (72.1%) and used river stream as house water source (73.2%).

Hygiene conditions were poor: 52.1% of participants did not have access to latrines and practiced open defecation. Surprisingly, a large part of the study population however declared having the opportunity to wash their hands (96.7%), using soap for washing hands (60.1%) and the habit of washing fruits or vegetables (59.3%) before eating/cooking.

More than half (51.6%) reported not having pigs near where they lived and pork consumption was reported by a high proportion (94.8%) of participants principally from district slaughter (64.7%). Most of the participants (78.9%) declared having a history of deworming drugs with a treatment taken at least twice a year. The survey data also showed that 69.3% of the study population was not aware of the concept of disease spreading within households. Nevertheless 60.1% of the people had knowledge about cysticercosis. Questionnaire analysis also

**Table 1. Characteristics, behaviours, health and hygiene conditions of the study population (n = 459).**

| | Number | % (95% CI) | | Number | % (95% CI) |
|---|---|---|---|---|---|
| **Gender** | | | **Habit of washing veggies** | | |
| Male | 235 | 51.2 (46.6–55.8) | Yes | 272 | 59.3 (54.7–63.7) |
| Female | 224 | 48.8 (44.2–53.4) | No | 80 | 17.4 (14.2–21.2) |
| Sex ratio | 1.05 | | Missing | 107 | 23.3 (19.7–27.4) |
| **Age groups (in years)** | | | **Pigs nearby** | | |
| Extreme (Median) | 5 – 82y (23.9y) | | Yes | 186 | 40.5 (36.1–45.1) |
| 5–10 y | 127 | 27.7 (23.8–32.0) | No | 237 | 51.6 (47.1–56.2) |
| 11–15 y | 75 | 16.3 (13.2–20.0) | Missing | 36 | 7.8 (5.7–10.7) |
| > 15 y | 257 | 56.0 (51.4–60.5) | **Pork consumption** | | |
| **Number of household members** | | | Yes | 435 | 94.8 (92.3–96.5) |
| 1 | 16 | 3.5 (2.1–5.6) | No | 14 | 3.1 (1.8–5.1) |
| 2 to 3 | 98 | 21.4 (17.8–25.4) | Missing | 30 | 6.5 (4.6–9.2) |
| 4 to 6 | 239 | 52.1 (47.5–56.6) | **Meat source (Pork)** | | |
| 7 to 9 | 94 | 20.5 (17.0–24.4) | Local/Home slaughter | 63 | 13.7 (10.9–17.2) |
| ≥10 | 12 | 2.6 (1.5–4.6) | District slaughter | 297 | 64.7 (60.2–68.9) |
| **Profession (> 15 y)** | | | Both | 69 | 15.0 (12.1–18.6) |
| Government official (Teacher) | 1 | 0.2 (0.0–1.5) | Missing | 10 | 2.2 (1.2–4.0) |
| Farmers | 250 | 54.5 (49.9–59.0) | **Concept of disease spreading within households** | | |
| Students | 145 | 31.6 (27.5–36.0) | Yes | 42 | 9.2 (6.8–12.2) |
| Missing | 21 | 4.6 (3.0–6.9) | No | 318 | 69.3 (64.9–73.3) |
| **Education** | | | Missing | 99 | 21.6 (18.0–25.6) |
| None (> 15y.) | 62 | 13.5 (10.7–17.0) | **Cysticercosis knowledge** | | |
| NA (Children ≤ 6y) | 42 | 9.5 (6.8–12.2) | Yes | 12 | 2.6 (1.5–4.5) |
| Elementary school | 315 | 68.6 (64.2–72.7) | No | 276 | 60.1 (55.6–64.5) |
| High school | 22 | 4.8 (3.2–7.2) | Missing | 171 | 37.7 (33.0–41.8) |
| College | 1 | 0.2 (0.0–1.5) | **Age of first seizure (year of old)** | | |
| Missing | 17 | 3.7 (2.3–5.9) | <2y. | 10 | 2.2 (1.2–4.0) |
| **House distance to the national main road** | | | 2-4y. | 15 | 3.3 (2.0–5.3) |
| < 6 km | 128 | 27.9 (24.0–32.2) | 5-10y. | 11 | 2.4 (1.3–4.2) |
| ≥ 6 km | 331 | 72.1 (67.8–76.0) | 11-15y. | 2 | 0.4 (0.1–1.6) |
| **House water source** | | | 16-19y. | 3 | 0.7 (0.2–1.9) |
| Stand pipe | 8 | 1.7 (0.9–3.4) | 20-37y. | 5 | 1.1 (0.5–2.5) |
| Stream (river) | 336 | 73.2 (69.0–77.0) | No | 63 | 13.7 (17.2–10.9) |
| Missing | 115 | 25.1 (21.3–29.2) | Missing | 350 | 76.3 (72.2–79.9) |
| **Latrine use** | | | **Headache frequency** | | |
| Yes | 201 | 43.8 (39.3–48.4) | 1 every 3 months or so (Rare) | 56 | 12.2 (9.5–15.5) |
| No | 239 | 52.1 (45.8–59.1) | > 1 every 3 months (Frequent) | 49 | 10.7 (8.2–13.9) |
| Missing | 19 | 4.1 (2.7–6.4) | Never | 257 | 56.0 (51.4–60.5) |
| **Opportunity for washing hands** | | | Missing | 97 | 21.1 (17.6–25.1) |
| Yes | 444 | 96.7 (94.6–98.0) | **Deworming frequency** | | |
| No | 1 | 0.2 (0.0–1.5) | Only when prescribed | 2 | 0.4 (0.1–1.7) |
| Missing | 14 | 3.1 (1.8–5.1) | 1/year | 60 | 13.1 (10.3–16.5) |
| **Soap use for washing hands** | | | 2/year | 362 | 78.9 (74.9–82.4) |
| Yes | 276 | 60.1 (55.6–64.5) | Never | 8 | 1.7 (0.9–3.5) |
| No | 127 | 27.7 (23.8–31.9) | Missing | 27 | 5.9 (4.1–8.5) |
| Missing | 56 | 12.2 (9.5–15.5) | | | |

Percentage (%) and 95% CI (95% Confidence Interval) are based on the total number of participants. NA = Not applicable.

showed that 56.0% of the participants declared never having headaches *versus* 10.7% declared having frequent headaches (more than 1 headache every 3 months). 10.1% of the participants reported having had their first seizure of which 1.1% (5 participants) were between 20–49 years old. However, a large proportion of the participants (76.3%) lacked data on their first seizure including 154 participants (31.6%) aged between 20–49 years old.

## Taeniosis prevalence

The estimated overall prevalence of taeniosis screened by stool microscopy for *Taenia* spp. carriage was 2.4% (95% CI: 1.3–4.3) corresponding to eleven tapeworm carriers living in eight distinct villages. Sedimentation technique detected 10 positive tapeworm carriers whereas Kato-Katz technique detected 9. *Taenia* carriage seems to be more prevalent in women (n = 7, 3.1%, 95% CI: 1.5–6.4) than men (n = 4, 1.7%, 95% CI: 0.6–4.5), and in adults (n = 7, 2.7%, 95% CI: 1.3–5.6) than children under 15 years old (n = 4, 2.0%, 95% CI: 0.7–5.2) but these differences were not statistically significant (P > 0.05). All age classes were affected: average age of the tapeworm carriers was 20.1 years, ranging from 6 to 51 years old. The number of detected tapeworm carriers varied slightly across villages: two for each of the villages of Kianjanomby, Sahavoemba, Torotosy and one for each of the villages Marojano, Mangevo, Ampitavanana, Ankazotsara, Ampitambe; No tapeworm carrier was detected in 4 out of the 12 villages where the sampling was carried out (Ambinanindranofotaka, Bevoahazo, Mandrivany and Sahavanana) (Table 2).

## *Taenia* species and *T. solium* genotypes

Of the 11 positive stools in microscopy, *cox1* PCR product was obtained from extracted copro-DNAs. Their sequencing analysis amplified *cox1* DNA fragments and showed that ten belonged to *T. solium* species and one was an unspecific amplification. Two *T. solium*

**Table 2. Prevalence of taeniosis by villages investigated.** Results obtained after stool analysis by Kato-Katz (KK) or sedimentation assays (SS).

| Villages investigated (stools analyzed/villagers >5y. old in the selected HH/ total villagers >5y. old[#], n) | Tapeworm carriers | | |
|---|---|---|---|
| | Positive samples by KK (n) | Positive samples by SS (n) | Positive samples per villages n (%, 95% CI) |
| Kianjanomby (50/89/721) | 1 | 2 | 2 (4, 1.0–14.9) |
| Sahavoemba (35/41/363) | 1 | 1 | 2* (5.7, 1.4–20.6) |
| Torotosy (52/80/383) | 2 | 2 | 2 (3.8, 0.9–14.3) |
| Marojano (23/50/303) | 1 | 1 | 1 (4.3, 0.6–26.2) |
| Mangevo (31/37/186) | 1 | 1 | 1 (3.2, 0.4–20.3) |
| Ampitavanana (40/65/432) | 1 | 1 | 1 (2.5, 0.3–16.1) |
| Ankazotsara (45/89/294) | 1 | 1 | 1 (2.2, 0.3–14.5) |
| Ampitambe (41/65/369) | 1 | 1 | 1 (2.4, 0.3–15.8) |
| Ambinanindranofotaka (40/48/253) | 0 | 0 | 0 |
| Bevoahazo (10/12/403) | 0 | 0 | 0 |
| Mandrivany (68/114/805) | 0 | 0 | 0 |
| Sahavanana (24/31/255) | 0 | 0 | 0 |
| **Total** | **9** | **10** | **11** |

n: number of villagers and participants, total and per village or positively detected by each test, prevalence (%) and 95% CI (Confidence Interval).

HH: household

[#]excluding 121 participants with unspecified age (not knowing their age and without birth certificate)

*Sahavoemba, two (2) different tapeworm carriers were detected either by KK or sedimentation coprology tests.

genotypes were identified: *T. solium* Asian genotype and *T. solium* African-American genotype. The majority of the sequences (9/10) belonged to the Asian genotype and was detected in stool samples from the villages of Ampitavanana, Torotosy, Ankazotsara, Kianjanomby, Mangevo, Ampitavanana and Sahavoemba. Only one *cox1* sequence matched with the African-American genotype and was identified from one of the tapeworm carriers living in the village of Sahavoemba (Fig 3).

The phylogenic tree analysis performed with *cox1* sequences, obtained from human stools, and the *cox1* Asian genotypes sequences (obtained from pig cisticerci) deposited in EMBL/GenBank databases revealed that the majority (8/9) of the sequences obtained in this study are absolutely conserved. These *T. solium* Asian genotypes clustered together within the corresponding genotype sequences isolated from pig cisticerci collected in Central, South and Western parts of Madagascar [40,41] and from other countries (India, Nepal, Japan and Thailand). All these *cox1* sequences belonged to the Asian *T. solium* genotype, cluster I (Fig 3). This first cluster was less close to the *cox1* DNA specimen collected from the eastern part of Madagascar (Toamasina, cluster II) and distant to those found in Indonesia (Asian *T. solium* genotype Cluster II) and in other countries such as Korea and China (Asian *T. solium* genotype Cluster IV). The last *cox1* *T. solium* Asian genotype sequence from Sahavoemba formed a cluster apart (Asian *T. solium* genotype Cluster V).

The only African-American genotype found in this study shared 100% similarity to the sequences already identified in the Southern part of Madagascar (Toliara, from pig cysticerci) and in other African/American countries (Tanzania and Mexico) forming a cluster (African-American *T. solium* genotype cluster III) which was less close to those found in Ecuador (African-American *T. solium* genotype cluster II) and more distant to those obtained in Mexico, Cameroon and Brazil (African-American *T. solium* genotype the Asian *T. solium* genotype cluster I). The variation of nucleotides and the corresponding amino acids for each *T. solium* genotype cluster are shown in the supporting S2 Fig.

## Exposure to *T. solium* parasite: seroprevalence of anti-*T solium* cysticercus antibodies (IgG)

The presence of anti-*T. solium* IgGs against metacestode glycoproteins in serum was estimated by ELISA (Ab-ELISA) and EITB methods, which given a fairly similar seroprevalence: 27.5% (95% CI: 23.5–31.7) and 29.9% (95% CI: 25.8–34.2) respectively (Table 3). Females seem to be more exposed to *T. solium* parasite using Ab-ELISA test (Sex ratio: 0.7) while no differential exposure was observed using the EITB assay (Sex ratio: 1.01). The average age of Ab-ELISA seropositive individuals was 22 years, (ranging from 5 to 79 years old) whereas the average age of EITB seropositive individuals was 24.4 years (ranging from 5 and 74 years old). Using both methods, anti-cysticercal IgG seroprevalence was quite similar between the three defined age-groups. Depending on the village investigated, the exposure rate found by Ab-ELISA and IgG Western Blot assays ranged between 13% - 50% and between 16% - 70% respectively (Table 3). At household level, the overall seroprevalence of IgG detected by Ab-ELISA and EITB within the same household in which a tapeworm carrier has been detected was 46% and 34% including 5/11 and 3/11 *Taenia* carriers respectively with co-infection (taeniosis and cysticercosis).

## Prevalence of current cysticercosis infection with viable cysts

Current cysticercosis was detected by measuring the excretory/secretory circulating *T. solium* antigens based on B158/B60 enzyme-linked immunosorbent assay (Ag-ELISA) [45]. Circulating antigens were detected in 57/459 participants giving an overall seroprevalence of 12.4% (95% CI: 9.7–15.8). Cysticercosis affected all age classes, ranging from 5 to 74 years old with an average age

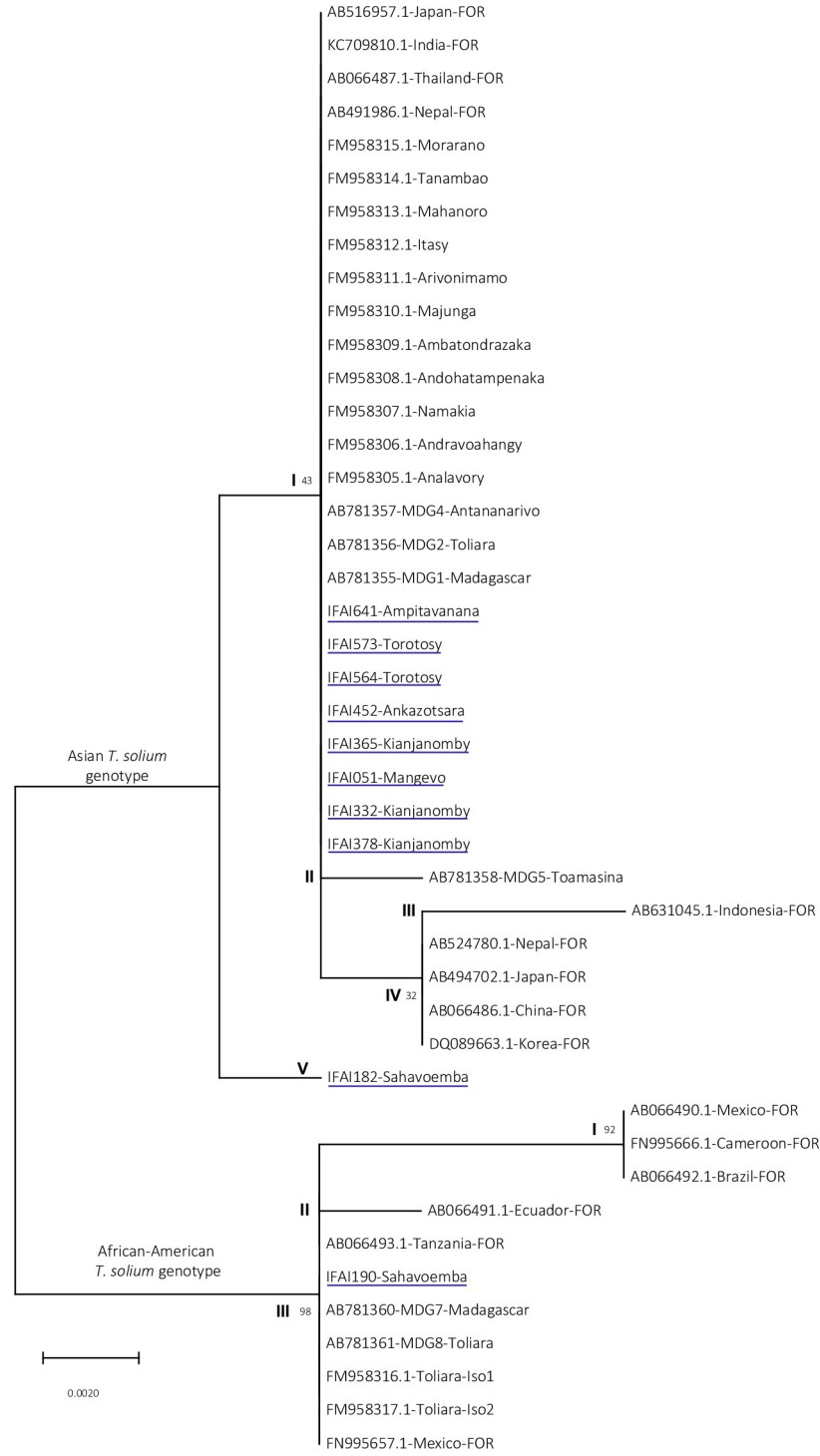

**Fig 3. Phylogenic tree showing the Asian and the African-American *Taenia solium* genotypes of the specimens obtained in this study and in databases.** *Cox1* sequences obtained in this study (from human stool *with* detected *Taenia* eggs) are underlined in blue. The sequences obtained from EMBL/Genbank databases (from cysticerci of pig) are indicated with their accession numbers followed by the name of each District/township for genes described in Madagascar and by "FOR" for specimens collected in foreign country. The percentage (%) of bootstrap replicates for the reliability tests of the associated taxa clustered together is shown next to the branch (up and down for the genomic and coding forms respectively).

**Table 3. Seroprevalence of *T. solium* anti-cysticercal antibodies (Ab-ELISA and EITB assays, IgG) and cysticercal Antigens (Ag-ELISA) for all participants investigated and within the *Taenia* carrier households.**

| | Ab-ELISA | | EITB | | Ag-ELISA | | |
|---|---|---|---|---|---|---|---|
| | Positive (n) | % (95% CI) | Positive (n) | % (95% CI) | Positive (n) | % (95% CI) | Doubtful n (%) |
| **Total, N = 459** | 126 | 27.5 (23.5–31.7) | 137 | 29.9 (25.8–34.2) | 57 | 12.4 (9.7–15.8) | 30 (6.5) |
| **Gender** | | | | | | | |
| Male | 53 | 22.6 (17.6–28.4) | 69 | 29.4 (23.9–35.5) | 36 | 15.3 (11.2–20.5) | 18 (7.7) |
| Female | 73 | 32.6 (26.7–39.0) | 68 | 30.4 (24.7–36.7) | 21 | 9.4 (6.2–14.0) | 12 (5.4) |
| **Sex ratio** | 0.7 | | 1.01 | | 1.7 | | 1.5 |
| **Age groups (in years)** | | | | | | | |
| Extreme (Median) | 5–79y (22y) | | 5–74y (24.4y) | | 5–74y (29.1y) | | 5–79 y (25.9) |
| 5–10y | 37 | 29.1 (21.9–37.7) | 34 | 26.8 (19.8–35.2) | 11 | 8.7 (4.8–15.0) | 7 (5.5) |
| 11–15y | 21 | 28.0 (19.0–39.3) | 25 | 33.3 (23.5–44.8) | 7 | 9.3 (4.5–18.4) | 5 (6.7) |
| > 15y | 68 | 26.5 (21.4–32.2) | 78 | 30.4 (25.0–36.3) | 39 | 15.2 (11.3–20.1) | 18 (7.0) |
| **Villages for all participants** (serum analyzed/villagers >5y. old in the selected HH/total villagers >5y. old[#], n) | | | | | | | |
| Kianjanomby (50/89/721) | 13 | 25 (15–39) | 23 | 44 (31–58) | 5 | 9.6 (4–21) | 0 |
| Sahavoemba (35/41/363) | 12 | 34 (20–52) | 10 | 29 (16–46) | 7 | 20.0 (10–37) | 4 (11) |
| Torotosy (52/80/383) | 10 | 19 (11–32) | 17 | 33 (21–47) | 11 | 21 (12–35) | 6 (12) |
| Marojano (23/50/303) | 5 | 22 (9–43) | 16 | 70 (48–85) | 2 | 9 (2–30) | 2 (9) |
| Mangevo (31/37/186) | 11 | 36 (21–54) | 9 | 29 (16–47) | 5 | 16 (7–34) | 4 (13) |
| Ampitavanana (40/65/432) | 14 | 35 (22–51) | 13 | 33 (20–4) | 8 | 20 (10–35) | 4 (10) |
| Ankazotsara (45/89/294) | 13 | 29 (17–44) | 8 | 178 (9–32) | 6 | 13 (6–27) | 2 (4) |
| Ampitambe (41/65/369) | 16 | 41 (27–57) | 9 | 23 (12–39) | 5 | 13 (5–28) | 2 (5) |
| Ambinanindranofotaka (40/48/253)* | 8 | 20 (10–35) | 10 | 25 (14–41) | 3 | 8 (2–21) | 1 (3) |
| Bevoahazo (10/12/403)* | 5 | 50 (21–79) | 2 | 20 (5–56) | 1 | 10 (1–50) | 2 (20) |
| Mandrivany (68/114/805)* | 16 | 24 (15–35) | 11 | 16 (9–27) | 2 | 3 (1–11) | 1 (2) |
| Sahavanana (24/31/255)* | 3 | 13 (4–33) | 9 | 38 (21–58) | 2 | 8 (2–29) | 2 (8) |
| **Village per *Taenia* carrier and per household** (mHH included/Total mHH) | | | | | | | |
| **Total, N = 41** | 19 | 46 (32–61) | 14 | 34 (22–50) | 8 | 20 (10–34) | |
| Kianjanomby (4/7) | 2[¤] | 50 (9–91) | 2[¤] | 50 (9–91) | 2[¤] | 50 (9–91) | 0 |
| Kianjanomby (2/4) | 0 | | 0 | | 0 | | 0 |
| Sahavoemba (6/8) | 2 | 33 (6–70) | 2 | 33 (6–70) | 1[¤] | 17 (1–56) | 0 |
| Sahavoemba (6/7) | 5[¤] | 83 (44–99) | 1 | 16 (1–56) | 1[¤] | 17 (1–56) | 0 |
| Torotosy (2/4) | 1[¤] | 50 (3–97) | 2[¤] | 100 (18–100) | 1 | 50 (9–91) | 0 |
| Torotosy (4/9) | 1 | 25 (1–70) | 1 | 25 (1–70) | 1 | 25 (1–70) | 0 |
| Marojano (3/6) | 1[¤] | 33 (2–88) | 1[¤] | 33 (2–88) | 0 | | 0 |
| Mangevo (4/5) | 2 | 50 (9–91) | 2 | 50 (9–91) | 0 | | 1 (25) |
| Ampitavanana (4/6) | 2 | 50 (9–91) | 2 | 50 (9–91) | 2 | 50 (9–91) | 0 |
| Ankazotsara (3/3) | 0 | | 0 | | 0 | | 0 |

(*Continued*)

**Table 3.** (Continued)

| | Ab-ELISA | | EITB | | Ag-ELISA | | |
|---|---|---|---|---|---|---|---|
| | Positive (n) | % (95% CI) | Positive (n) | % (95% CI) | Positive (n) | % (95% CI) | Doubtful n (%) |
| Ampitambe (3/6) | 3[¤] | 100 (44–100) | 1 | 33 (2–88) | 0 | | 1 (33) |

Ab-ELISA: LLGP antigens-based enzyme-linked immunosorbent assay.

EITB: Enzyme-linked Immuno-electroTransfer Blot detecting specific antibodies (IgG) against *T. solium* cysticercus glycoproteins (GP39/42-GP24-GP21-GP18-GP14/13 kiloDaltons, LLGP antigens) [44].

Ag-ELISA: B158/B60 monoclonal antibodies-based enzyme-linked immunosorbent assay. An Ag-index less than 0.8 was classified as negative result, an Ag-index greater than or equal to 1.3 was considered as positive and values in between classified as doubtful.

n: number of villagers and participants, total and per village or detected positively by each serological test respectively

HH or mHH: household or member of household including participant tested positively

[#]excluding 121 villagers with unspecified age (those not knowing their age and without birth certificates)

[*]Villages where no *Taenia* eggs carriers were detected.

[¤]*Taenia* carrier tested positively for each serological test

of 29.1 years (Table 3). *T. solium* antigen seroprevalences were higher in males (15.3%, sex ratio: 1.7) and in adults over 15 years old (15.2%) but did not reach statistical significance (p = 0.07 and p = 0.32 respectively). The seroprevalence of *T. solium* antigens varied between 3% and 21% according to the village investigated. Doubtful cases (n = 30, 6.5%, 95% CI: 4.6–9.2), with an Ag-index < 0.8, were also detected in the study population. These doubtful cases were higher in men (7.7%, sex ratio: 1.5) and quite similar between the 3 age groups. The rate of these cases, which should be confirmed by a second blood sampling, varied from 0% to 20% according to the village investigated (Table 3). Considering the participants within the same household, the overall prevalence of circulating *T. solium* antigens was 20% comprising 3/11 *Taenia* carriers.

## Risk factors associated with *T. solium* infections: univariate analysis

Several epidemiological and behavioral factors were evaluated for their association with *T. solium* infections and summarized in Table 4. For both serological techniques used (EITB and Ag-ELISA), sixteen variables were firstly assessed using univariate analysis. Univariate analysis suggests that a high number of householders (≥ 4 people, OR = 2.0, p = 0.001) and open defection behaviors (OR = 1.6, p = 0.02) could be risk factors associated with the presence of *T. solium* anti-cysticercal antibodies (EITB). With p value ≤ 0.2, housing more than 6 km from the national main road (OR = 1.5), pork consummation (OR = 3.1), district slaughter as meat (pork) source (OR = 0.6) and frequent headache (OR = 0.6) were also included in the multivariate analysis for EITB test (Table 4). Contrariwise, regarding *T. solium* circulating antigen positivity (Ag-ELISA seropositivity), being male (OR = 0.6 for female, p<0.001) and no use of latrine (OR = 1.6, p = 0.049) could be associated with a higher risk of developing active cysticercosis. Multivariable analysis was completed for Ag-ELISA test with the following factors (with p value ≤ 0.2): adulthood (over 15 years, OR = 1.7), use of stand pipe for house water source (OR = 0.3 for stream river), use of soap for washing hands (OR = 1.5), and no habit of washing veggies (OR = 0.7 for habit of washing veggies).

## Risk factors associated with *T. solium* infections: multivariate analysis

In the multiple regression analysis, the risk factors that were significantly associated with anti-cysticercal Antibodies (EITB) or cysticercal Antigens (Ag-ELISA) positivity (p-value ≤ 0.05)

**Table 4. Univariate analysis on risk factors and *T. solium* infections (positive EITB and Ag-ELISA results).**

| Variables | Positive EITB | | Positive Ag-ELISA | |
|---|---|---|---|---|
| | OR (95% CI) | *p-value* | OR (95% CI) | *p-value* |
| **Age groups (in years)** | | | | |
| 5–10y[a] | 1 | | 1 | |
| 11–15y | 1.4 (0.7–2.5) | 0.323 | 1.1 (0.5–2.6) | 0.700 |
| > 15y | 1.2 (0.7–1.9) | 0.468 | 1.7 (1.0–3.1) | 0.100* |
| **Gender** | | | | |
| Male[a] | 1 | | 1 | |
| Female | 1.0 (0.7–1.6) | 0.816 | 0.6 (0.4–0.9) | 0.000* |
| **Number of household members** | | | | |
| ≤3 persons[a] | 1 | | 1 | |
| ≥ 4 persons | 2.0 (1.2–3.2) | 0.001* | 1.0 (0.6–1.8) | 0.900 |
| **Education** | | | | |
| No schooling[a] | 1 | | 1 | |
| Elementary and high school levels | 1.4 (0.8–2.2) | 0.220 | 0.8 (0.5–1.4) | 0.516 |
| **House distance to the national main road** | | | | |
| < 6 km[a] | 1 | | 1 | |
| ≥ 6 km | 1.5 (1.0–2.3) | 0.070* | 1.0 (0.6–1.7) | 0.940 |
| **House water source** | | | | |
| Stand pipe[a] | 1 | | 1 | |
| Stream (river) | 1.2 (0.3–6.2) | 0.800 | 0.3 (0.1–1.2) | 0.090* |
| **Washing hands after toilet, before eating** | | | | |
| Yes[a] | 1 | | 1 | |
| No | 1.1 (0.6–2.1) | 0.744 | 0.6 (0.3–1.5) | 0.260 |
| **Soap use for washing hands** | | | | |
| No[a] | 1 | | 1 | |
| Yes | 0.9 (0.6–1.4) | 0.633 | 1.5 (0.8–2.6) | 0.179* |
| **Habit of washing veggies** | | | | |
| No[a] | 1 | | 1 | |
| Yes | 1.1 (0.8–1.7) | 0.559 | 0.7 (0.5–1.2) | 0.179* |
| **Pigs nearby** | | | | |
| No[a] | 1 | | 1 | |
| Yes | 0.9 (0.6–1.4) | 0.753 | 1.1 (0.7–1.8) | 0.672 |
| **Pork consumption** | | | | |
| No, without response[a] | 1 | | 1 | |
| Yes | 3.1 (0.9–10.6) | 0.070* | 0.9 (0.3–2.4) | 0.809 |
| **Meat source (Pork)** | | | | |
| Local/Home slaughter or both[a] | 1 | | 1 | |
| District slaughter | 0.6 (0.3–1.1) | 0.070* | 0.8 (0.5–1.3) | 0.412 |
| **Latrine use** | | | | |
| Yes[a] | 1 | | 1 | |
| No | 1.6 (1.1–2.4) | 0.020* | 1.6 (1–2.6) | 0.049* |
| **Disease spreading concept within households** | | | | |
| No/without response[a] | 1 | | 1 | |
| Yes | 1.2 (0.6–2.4) | 0.610 | 0.8 (0.4–2.0) | 0.692 |
| **Frequent headache (> 1 every 3 months)** | | | | |
| No[a] | 1 | | 1 | |
| Yes | 0.6 (0.3–1.2) | 0.131* | 2.0 (0.5–2.1) | 0.912 |

(*Continued*)

**Table 4.** (Continued)

| | Positive EITB | | Positive Ag-ELISA | |
|---|---|---|---|---|
| **Variables** | **OR (95% CI)** | **p-value** | **OR (95% CI)** | **p-value** |
| **Deworming frequency (2/year)** | | | | |
| No[a] | 1 | | 1 | |
| Yes | 1.3 (0.8–2.1) | 0.324 | 1.5 (0.8–2.8) | 0.203 |

[a] Reference variable

*Risk factors with a p value ≤ 0.2 were included in multivariable analysis.

are presented in Tables 5 and 6 respectively. A high number of householders (≥ 4 people, OR = 1.9, 95% CI: 1.1–3.1, p = 0.017) and open defecation behaviors (OR = 1.5, 95% CI: 1.0–2.3, p = 0.038) were found to be the main risk factors of exposure to *T. solium* infections (sero-positivity for *T. solium* antibodies) [see Table 5]. On the other hand, females (OR = 0.5, 95% CI: 0.3–0.9, p = 0.010) and house water source from river (OR = 0.4, 95% CI: 0.2–0.7, p<0.001) were less at risk of developing active cysticercosis with the presence of circulating *T. solium* antigens (Table 6). At the opposite, adults (over 15 years old, OR = 1.6, 95% CI: 1.0–2.7, p = 0.049) had an increased risk of developing cysticercosis with detectable *T. solium* circulating Ag (Table 6).

## Discussion

There is little information available on taeniosis/cysticercosis epidemiology in Madagascar [21]. To the best of our knowledge, the current study is the first community-based survey reporting the prevalence of human *T. solium* infections using three different diagnostic tests for cysticercosis and analyzing the associated risk factors in rural and remotely region of Madagascar.

Little data is available regarding the prevalence of adult tapeworm carriers (taeniosis) who are the source of cysticercosis in humans and pigs [54]. The main difficulty for studying taeniosis especially in Africa is the lack of a simple, sensitive and *T. solium* specific diagnostic tool. Stool microscopy techniques classically used are not able to discriminate eggs of *T. solium* and *T. saginata* and have poor sensitivity and specificity [55]. In Madagascar, national surveys did not report any tapeworm carriers in the district of Ifanadiana [35]. Kato Katz and Spontaneous Sedimentation techniques used in this study however allowed estimating a high overall prevalence of taeniosis reaching 2.4% and classifying these rural areas as hyperendemic [56] and at a higher rate when compared to the results reported previously in Madagascar [35,57]. The prevalence of taeniosis found in this study are quite similar to those previously described (0.1%–1.4%) in many African countries (Burundi, Congo, Ethiopia, Togo, Zambia, Cameroon, Kenya and Tanzania) and in some endemic countries of Latin America (Peru and Ecuador) using comparable coprological examination. Conversely these prevalence rates could be low compared to those reported in other African countries (Guinea, Nigeria, Ghana, and Gambia: up to 13%) [12,14,15,54] and those obtained using more sensitive diagnostic tools such as coproantigen ELISA and EITB methods (Zambia, Tanzania, Kenya and Democratic Republic of Congo: up to 23.4% [58–61].

No adult tapeworm carriers were detected in 4 out of the 12 villages investigated where participants presenting antibodies anti-*Taenia solium* and circulating antigens were however detected. Indeed, studies have shown that microscopy techniques are weakly sensitive missing 60–70% cases of taeniosis [62]. The best diagnostic assay for the intestinal taeniosis and

**Table 5. Multivariable analysis on risk factors for exposure to *T. solium* infections (positive EITB).**

| | Positive EITB | | | |
|---|---|---|---|---|
| | Initial model | | Final model | |
| **Variables** | **OR (95% CI)** | **p-value** | **OR (95% CI)** | **p-value** |
| **Number of household members** | | | | |
| ≥ 4 persons | 1.7 (1.0–3.0) | 0.047* | 1.9 (1.1–3.1) | 0.017* |
| **Education** | | | | |
| Elementary and high school levels | 1.6 (0.9–2.6) | 0.088 | | |
| **House distance to the national main road** | | | | |
| ≥ 6 km | 1.4 (0.9–2.2) | 0.181 | | |
| **Pork consumption** | | | | |
| Yes | 3.0 (0.8–10.6) | 0.097 | | |
| **Meat source (Pork)** | | | | |
| District slaughter | 0.8 (0.5–1.2) | 0.267 | | |
| **Latrine use** | | | | |
| No | 1.6 (1.0–2.4) | 0.038* | 1.5 (1.0–2.3) | 0.038* |

OR: odds ratio; 95% CI: 95% confidence interval.

*Significance level: $p \leq 0.05$.

constituting an effective tool for epidemiological studies would be the coproantigen detection ELISA having a sensitivity of about 95% and a specificity over 99% [62]. However these data have never been independently validated. This ELISA assay has been estimated to be two to ten times more sensitive than coprology methods [63]. But neither the corresponding test nor the necessary polyclonal antibodies for the coproantigen ELISA are available on the market and the method remains genus specific. Regarding molecular tools, a triplex Taq-Man probe-based qPCR for the detection and discrimination of *T.solium*, *T. saginata* and *T. asiatica* in human stool [64] and the field application of a Loop-mediated isothermal AMPlification method (LAMP) for rapid identification of human taeniosis were also reported [65]. Using

**Table 6. Multivariable analysis between risk factors and *T. solium* infections (positive Ag-ELISA).**

| | Positive Ag-ELISA | | | |
|---|---|---|---|---|
| | Initial model | | Final model | |
| **Variables** | **OR (95% CI)** | **p-value** | **OR (95% CI)** | **p-value** |
| **Age classes** | | | | |
| > 15y | 1.8 (1.1–3.1) | 0.026* | 1.6 (1.0–2.7) | 0.049* |
| **Gender** | | | | |
| Female | 0.5 (0.3–0.8) | 0.007* | 0.5 (0.3–0.9) | 0.010* |
| House water source | | | | |
| Stream (river) | 0.3 (0.1–1.5) | 0.100 | 0.4 (0.2–0.7) | 0.000 * |
| **Soap use for washing hands** | | | | |
| Yes | 1.2 (0.7–2.2) | 0.467 | | |
| **Habit of washing veggies** | | | | |
| Yes | 0.7 (0.4–1.1) | 0.139 | | |
| **Toilet use** | | | | |
| No | 1.6 (1.0–2.7) | 0.057 | | |

OR: odds ratio; 95% CI: 95% confidence interval

Significance level: $p \leq 0.05$*

these tests especially the simple and sensitive LAMP technique in the context of Madagascar could be very useful in the field.

Taeniosis cases reported in this study affected all age (mean age: 20.1 years) with 2.7% of prevalence in adults more than 15 years old. This population is not targeted by the national mass treatment programs (mainly treating children schooled between 5 to 15 years old). Thereby these results could guide the national program on mass chemoprevention (as primary intervention strategy against *T. solium* infections) to treat the entire population of all ages, especially the inhabitants of rural and remote areas.

Species of the *Taenia* spp. eggs identified in this study were all *T. solium*, the only species associated with cysticercosis. In order to analyze *T. solium* genotypes, we used cytochrome c oxidase subunit I which has been widely used for studying polymorphism and for establishing phylogenetic trees of *Taenia* genus (*T. solium*, *T. saginata* and *T. asiatica*) [41]. Our results confirm the findings of previous molecular analysis of *T. solium* worldwide, reporting that *T. solium* could be divided into two genotypes, Asian and African/American genotypes [66–68] (Fig 3). Our data strengthen the previous results of Michelet *et al.* and Yanagida *et al.* [40,41] showing that the two genotypes geographically disjunct are sympatric in Madagascar. Indeed confirmed by analyses of mtDNA and nuclear DNA, Malagasy people have mixed origins from Southeast Asia and from Southeast Africa [69]. Our results extend the previous findings that were made in Madagascar (on cysticerci from pigs) that the sympatric distribution of Asian and African-American *T. solium* genotypes is also found in the adult form and in the eggs of *T. solium* from human stool. The predominance of the Asian genotype (9/10 specimens) corroborated also to these prior results observed at a national level which occurred evenly at a District level of Ifanadiana for this study (Fig 3). Asian genotypes from this study are close to the Asian genotypes from India and Thailand but distant to the Asian genotypes from Indonesia, Korea and China as already reported [40,41], supporting the importance of the Indian influence on the diversity of people and culture in Madagascar [40,69,70]. In this study, one of the two sequences obtained from the village of Sahavoemba resembles the Asian genotype but contains a base at the position 867 that is substituted (adenosine to guanine), as found in the African-American genotypes (Fig 3). Cross-fertilization and hybridization between individual worms possessing different genotypes has already been suggested by analyzing the nuclear genes [40]. Only one specimen from the village of Sahavoemba was an African-American genotype in our study. The sequence is closer to the other specimens isolated in Madagascar (Toliara) and Tanzania but distant to the sequences from Cameroon and Brazil (Fig 3).

The antigens used in this study based on lentil lectin affinity chromatography preparation [44] and Ag-ELISA (B158/B60) method [45] are used worldwide and constitute assays recognized by the WHO, both in sero-surveys for prevalence estimation and also for the diagnosis of individuals [50].

In this study the prevalence of antibodies (IgG) associated with an exposure with *T. solium* parasite detected by ELISA and by EITB methods using *T. solium* metacestode glycoproteins was quite similar (27.5% and 29.9% respectively). One study in Zambia has reported a higher seroprevalence (34.5%) than that found in this study using an EITB assay [71]. However this positivity rate is relatively high compared to those found in the general population or among villagers in other African countries using similar methods (ranging from 1.3% to 14.3%) and very high compared to the prevalence obtained in people with epilepsy (0 to 2.8%) [12–15,54]. The detection of cysticercosis-specific antibodies during field-based studies could help to identify areas of disease transmission [50] however they tend to over-estimate the prevalence of cysticercosis in sero-epidemiological surveys. Indeed, as antibodies disappear after 1 to 3 years in only 30–40% of seropositive people in endemic countries [72], some seropositivity will

reflect past exposure to the parasite. In addition, infection in the general population could be in any tissue unlike epileptic individuals with NCC with cysts mainly localized in the Central Nervous System. Lower antibodies related to NCC may also be due to the lower sensitivity of EITB in parenchymal localization with few or calcified cysts [73] [74].

The presence of circulating antigens asserting an active cysticercosis case with viable cysts was observed at a prevalence of 12.4%. This prevalence is relatively high compared to those obtained in the following African countries (Cameroon, Zambia, Kenya, Burkina-Faso, Togo, and Senegal: 0.7% to 8.1%) [12–14,16,58] but extensively low compared to those found in Tanzania, Congo, and Togo (16.8% à 38%) [12,13]. However this positivity is similar to the prevalence of circulating *Taenia* antigens detected in Zambia, 12.5%[71].

Regarding the associated risk factors analysis, the use of soap for hand-washing was surprisingly found associated to active cysticercosis. A study using the Ag-ELISA test reported that participants who declared washing their hands by a dipping method and using the same water were more likely to be seropositive unlike those who used running water [59]. In our study villages, using soap for hand washing as a risk factor may be explained by the fact that people use the same contaminated river water for multiple purposes (washing vegetables, dishes, hands, etc). However other factors would be interesting to consider and could help understand this surprising result. The risk analysis result obtained with the variable "house water source" using standpipe and the probable protective factor when using the river with renewed running water seems to strengthen this previous report. In fact, clean water used directly as a source of water might be a protective factor of cysticercosis in this study.

Other risk factors tested significantly in this study are common factors classically reported in several previous investigations of *T. solium* infections in Africa and worldwide, such as open defecation [54,71,75–78], pork consumption (especially in preparation and cooking methods) [54,77,79] and increased age (adults >15 year old) [2,12,59,80–83]. Many family members (≥4) in the same household is a high risk insofar as a single *T. solium* carrier may easily cause cysticercosis to his family members through *T. solium* eggs shedding [11,84,85]. Among all identified factors, lack of knowledge of the population about soil-transmitted diseases and their risk (68.8% of the participants with elementary level of education) [16,75,86], the poor hygiene (52.1% of participants without latrines and practicing open defecation) and the isolation of these villages (≥ 6km from the main national road) promote the spread of *T. solium* [17,87]. Therefore, public health education in Madagascar should be a complementary approach for *T. solium* control. Indeed, these education and communication-based strategies have already been evaluated with success in several places like in rural communities of Burkina Faso [88], in an urban community of Mexico [54], through a computer-based educational health tool on *T. solium* in Tanzania and in Zambia [89,90], or also through a participatory community education workshop about *T. solium* in Peru [91]. Some protecting factors have nonetheless been identified as being a female, district *versus* local slaughter for pork source and habit of washing vegetables. Similar results were also obtained in village-community studies performed in Congo and in Burkina Faso that have reported a positive association between the presence of *T. solium* circulating antigens and being a male [2,61,92]. Likewise, several publications have reported that in the Districts where official meat inspection could take place, unlike local/home unregistered slaughtering (that has been largely shown to be an important source of transmission of *T. solium* infections), the prevalence of parasite carriage and associated infections decreased significantly [15,17,93–95].

Our study may present limitations related to the number of samples analyzed. However, the epidemiological data obtained on the geographical distribution of *T. solium* taeniosis / cysticercosis in humans living in remote areas of Madagascar is important for identifying high-risk populations and applying control measures. This study identified and treated some tapeworm

carriers of all age. Indeed, the treatment of taeniosis in humans is one of the basic "quick impact" interventions for strengthening the control/elimination of the parasite transmission and preventing the health burden associated with NCC. During this study, eight cases of active cysticercosis including three co-infections (taeniosis and active cysticercosis probably through self-infestation) were observed and clustered at household level with *Taenia* carrier. Some suggestive clinical manifestations of health burden associated with NCC (seizure in individuals aged between 20–49 years old [96]) were also recorded without being significantly associated with cysticercosis (Probably due to the high number of missing data: 76.3%) and without confirmation of diagnosis by neuroimaging. Indeed, neuroimaging is an Absolute-Major-Confirmative criteria of NCC. Nevertheless, important criteria of NCC according to the classification of Del Bruto and colleagues in 2017 (such as: specific anticysticercal antibodies/cysticercal antigens detected by immunodiagnostic tests, evidence of a household contact with *T. solium* infection, and clinical manifestations suggestive of NCC: mainly seizures) were recorded during this study. These data were reported to the local health authorities for the confirmation of the diagnosis of these cases and their eventual management.

Human health interventions in Madagascar remain challenging. Indeed, through a three-year pilot project (2015–2017) in Madagascar, a significant reduction in the prevalence of taeniosis was observed after annual Mass Drug Administration (MDA) targeted adults and children >5 years old. However this reduction could not be maintained [97]. Investigations in pig health and environmental sector as part of a "one health" approach are therefore required to break the cycle of infection [98]. An ongoing project combining pig vaccination, drug treatment in pigs and in human tapeworm carrier will investigate this approach in Madagascar [99].

## Conclusions

This is the first large-scale study in Madagascar to examine the prevalence of *Taenia solium* taeniosis/cysticercosis in humans and the association between potential risk factors measured at the individual-, household-, and village-level. Our results showed high rates of taeniosis cases affecting all ages in both townships investigated, strongly exposing human and pigs to cysticercosis by considering open defecation practices of the participants. Indeed, search of *Taenia* carrier among household contacts is currently recommended to identify the potential source of infection clustered within household and to reduce further spread/burden of the diseases. This high exposure of participants confirmed by the high rate of active cysticercosis cases should be managed with neuroimaging for a targeted and more efficient treatment. These data would be useful for guiding the authorities and all the entities involved in the programs and in the strategies for combating these diseases especially in rural and remote areas in Madagascar Island. Investigations in pigs and in environment would also be necessary as part of a "one health" approach to complete control strategies for those zoonotic neglected tropical diseases.

## Supporting information

**S1 Fig. Position of the common primers used to amplify *cox1* gene and discriminate *Taenia* species by PCR and sequencing.** Multiple alignments of *T. solium* Asian or African/American genotypes (*T. sol* Gen Asia, GeneBank Accession No˚ AB066488; and *T. sol* Gen Af/Am, GeneBank Accession No˚ AB066492.1), *T. asiatica* (GeneBank Accession No˚ AB107235.1) and *T. saginata* (GeneBank Accession No˚ AB107246.1) were performed. Similarity and variation are marked by a star and space respectively. Common forward and reverse primers are

indicated highlighted in bold and with an arrow.
(DOCX)

**S2 Fig. Multiple alignment of the partial nucleotide sequence of *cox1* showing polymorphism between *T. solium* genotypes obtained in this study and in databases.** Five *T. solium* Asian genotypes (*T. sol* Asian I to V) and three for *T. solium* African-American genotype *(T. sol* Af/Am I to III) were aligned. *T. solium* Asian genotype cluster I sequence is shown as reference. Similarity is indicated with dot. Variant nucleotides are shown in lowercase and in capital letters if affecting amino acid sequence. Comparing to the Asian *T. solium* genotype cluster I (containing the majority of sequences obtained in this study), the Asian genotype cluster II (from Toamasina, Madagascar) showed one changed base (G instead of A) at the position 934 of the *cox 1* completed sequence which changed the corresponding amino acid while the Asian genotype cluster III (from Indonesia) presented 3 substituted nucleotides (C, G and A changed into T, A and G: at the positions 650, 666 and 994 respectively). This last changed base was also the only variation observed in the Asian genotype cluster IV (from Nepal, Japan, China and Korea). The last Asian *T. solium* genotype obtained in this study, from Sahavoemba (Asian genotype cluster V) was also closely related to the majority of Asian *T. solium* genotype found in this study (Cluster I) except for one base: A modified in G at the position 867 which is a common substitution in the African-American *T. solium* genotype. The only African/American genotype found in this study forming the African-American genotype cluster III (with Tanzania, Toliara Madagascar and Mexico) showed six substituted nucleotides (of which 3 modified amino acids) compared to the Asian genotype cluster II. The African-American genotype cluster II (Ecuador) and I (Mexico, Cameroon and Brazil) counted seven and nine substitutions respectively both with four changed amino acids.
(TIF)

**S1 Table. Composition of the Townships and Fokontany (Ifanadiana District) investigated in this study.** The geographic coordinates (GPS) of the 12 villages are indicated.
(DOCX)

## Acknowledgments

We are grateful to all communities involved in this study: the villagers, the local authority, the community workers and facilitators. We would like to acknowledge the Centre ValBio (CVB) team and the CVB health team: Maya Moore, Pascal Rabeson, Jesse McKinney, Rakotoarison M. Fara Nantenaina, Andry Andriamidanarivo, Fara Maria Violette Nambinintsoa and Francis Daniel Lovasoa for field logistics and overall support. Our most heartfelt thanks to our colleagues at IPM: Nônô Randrianasolo, Mamy Donah Andrianatoandro and Fanirisoa Randrianarisaona (for technical help), Jean Marius Rakotondramanga, Anjasoa Randrianarijaona and Astrid Knoblauch (for statistical advices), Mireille Harimalala (for phylogeny help), Thiery Nirina Jean Jose Nepomichene (for the bibliography part) and Henintsoa Malalatiana TIDA (for the English review). We are grateful to the neuropsychiatry department of Befelatanana Hospital for the serum samples used as controls. We also thank MICET (Malagasy Institute for the Conservation of Tropical Environments) team for their logistical support.

## Author Contributions

**Conceptualization:** Anjanirina Rahantamalala, Luis A. Marcos, Inès Vigan-Womas.

**Data curation:** Anjanirina Rahantamalala, Rado Lalaina Rakotoarison, Emma Rakotomalala, Mahenintsoa Rakotondrazaka, Jaydon Kiernan, Paul M. Castle, Lee Hakami, Koeun Choi, Luis A. Marcos, Inès Vigan-Womas.

**Formal analysis:** Anjanirina Rahantamalala, Aina Harimanana, Inès Vigan-Womas.

**Funding acquisition:** Patricia Wright, Peter M. Small, Luis A. Marcos, Inès Vigan-Womas.

**Investigation:** Anjanirina Rahantamalala, Rado Lalaina Rakotoarison, Emma Rakotomalala, Mahenintsoa Rakotondrazaka, Jaydon Kiernan, Paul M. Castle, Lee Hakami, Koeun Choi, Luis A. Marcos.

**Methodology:** Anjanirina Rahantamalala, Armand Solofoniaina Rafalimanantsoa, Simon Grandjean Lapierre, Luis A. Marcos, Inès Vigan-Womas.

**Project administration:** Anjanirina Rahantamalala, Luis A. Marcos, Inès Vigan-Womas.

**Resources:** Rado Lalaina Rakotoarison, Emma Rakotomalala, Mahenintsoa Rakotondrazaka, Armand Solofoniaina Rafalimanantsoa, Inès Vigan-Womas.

**Supervision:** Anjanirina Rahantamalala, Luis A. Marcos, Inès Vigan-Womas.

**Validation:** Anjanirina Rahantamalala, Luis A. Marcos, Inès Vigan-Womas.

**Visualization:** Anjanirina Rahantamalala, Inès Vigan-Womas.

**Writing – original draft:** Anjanirina Rahantamalala, Inès Vigan-Womas.

**Writing – review & editing:** Anjanirina Rahantamalala, Patricia Wright, Simon Grandjean Lapierre, Matthieu Schoenhals, Peter M. Small, Luis A. Marcos, Inès Vigan-Womas.

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
