## [Decision Letter · Decision Letter 0]

29 Jun 2021

Dear Dr Rahantamalala,

Thank you very much for submitting your manuscript "Prevalence and factors associated with human Taenia solium taeniasis and cysticercosis in twelve remote villages of Ranomafana rainforest, Madagascar" for consideration at PLOS Neglected Tropical Diseases. As with all papers reviewed by the journal, your manuscript was reviewed by members of the editorial board and by several independent reviewers. In light of the reviews (below this email), we would like to invite the resubmission of a significantly-revised version that takes into account the reviewers' comments. 

We cannot make any decision about publication until we have seen the revised manuscript and your response to the reviewers' comments. Your revised manuscript is also likely to be sent to reviewers for further evaluation.

Sincerely,

Jessica K Fairley, MD, MPH

Associate Editor

Aaron Jex

Deputy Editor

Reviewer's Responses to Questions

**Key Review Criteria Required for Acceptance?**

**Methods**

-Are the objectives of the study clearly articulated with a clear testable hypothesis stated?

-Is the study design appropriate to address the stated objectives?

-Is the population clearly described and appropriate for the hypothesis being tested?

-Is the sample size sufficient to ensure adequate power to address the hypothesis being tested?

-Were correct statistical analysis used to support conclusions?

-Are there concerns about ethical or regulatory requirements being met?

Reviewer #1: The objectives are clearly stated and the study design seems appropriate, but some important details are still missing (see below). In general, the analyses seem appropriate and ethical and regulatory requirements are met.

L157: how exactly were the households randomized? And out of how many households were these households randomly selected (i.e. how many were not selected)? And why 164? A sample size consideration is missing. 

L196: please give an indication of the time of conservation (between stool sample collection and extraction/PCR analysis)

L188-193: The questionnaire should be described more clearly. The results of the questions could be very different, depending on how the questions were formulated. For example the question pork consumption (yes/no) is too vague. It could be interpreted as having ever eaten pork in your life, but also as eating pork weakly, which are two completely different things. I would suggest to add the questionnaire in appendix.

Is having more than once headache every 3 months considered as chronic headache? It would be good to add a reference to this definition of chronic headache.

Reviewer #2: The hypothesis is clearly stated as the determination of the prevalence of human taenia solium, taeniasis/cysticercosis and risk factors in remote areas of Madagascar. The human population of the area is clearly described. The study design is appropriated and the sample size seems sufficient to address this hypothesis, statistical analyses do not rise any concern. Ethical and regulatory requirement seem to have been met.

Reviewer #3: The objectives of the study are clearly articulated and the population is also clearly described. The sample size is sufficient to ensure adequate power to address the hypothesis. The statistical analysis support conclusions.

**Results**

-Does the analysis presented match the analysis plan?

-Are the results clearly and completely presented?

-Are the figures (Tables, Images) of sufficient quality for clarity?

Reviewer #1: The analyses match the analysis plan and the results are compete, but at times the results are over interpreted (especially regarding the results at village level). 

The primary aim and secondary/exploratory outcomes could be defined more clearly. I think the sample size is not appropriate for a thorough risk factor analysis. In my opinion, the prevalence estimates seem like the primary aim, and the risk factors are more exploratory. I would suggest to make that more clear. 

L348: 574 willing participants, out of how many in these households? Do you have an indication how many people in the selected households were not willing to participate? And L350: not really out of 574 villagers, but out of 574 participants. I assume that the number of villagers is higher than this number because of the exclusion criteria (above 5) and willingness to participate. If available, it would be good to provide this information in Figure 2 and/or the text (how many were excluded because of age or because they were not willing).

L389: this is an over-interpretation of the data; I don’t think you should conclude these villages are “most infected” based on the data (because it’s based on 0, 1 or 2 positive cases per village). This is also clear from the wide CI around the estimates in Table 2. The % positive samples per village should therefore not have digits, because the estimates are very uncertain due to the relatively low sample size per village. It’s OK to present the results per village, but don’t overanalyse them. As mentioned earlier, it would be good to give an indication of the number of people living in each village, because that’s not clear from the manuscript. For example the results from only 10 people from Bevoahazo: how many people actually lived in that village at the time of recruitment? 

L460-463: I find this sentence very difficult to follow. And also for the sentence below: the message is not very clear, and as mentioned above, the categorization of villages between presence/absence of taeniosis based on the very few positive samples is likely an overinterpretation of the results due to the large uncertainty around the taeniosis prevalence estimates per village.

L466: the highest prevalence in Marojano: the sample size is too low to draw conclusions about which village has the highest prevalence (Marojano is only based on 23 samples)

L485-489: same comment as above

L500: how was this cutoff (4 or more) determined? Same comment for house distance. 

Figure 2: first box with Excluded: “missing epidemiological data”: what exactly does this mean? In table 1, it seems there are missing data for some of the 459 participants (but they are not excluded)? It would be better to specify how many are missing due to missing epidemiological data (and what exactly is meant) and how many due to missing biological samples (and which: serum and/or stool). Second box with Excluded: all serological analyses (not analysis)

Reviewer #2: Analyses presented match the analysis plan and the results are clearly presented.

Reviewer #3: The results and figures are clearly and completely presented. The figures are of sufficient quality for clarity.

**Conclusions**

-Are the conclusions supported by the data presented?

-Are the limitations of analysis clearly described?

-Do the authors discuss how these data can be helpful to advance our understanding of the topic under study?

-Is public health relevance addressed?

Reviewer #1: The discussion clearly addresses the public health relevance and also includes some limitations of the study. The discussion could be slightly more comprehensive.

L540-550: this whole section seems rather introduction than discussion. The results from the current manuscript are currently not mentioned/discussed here. 

L582: “more prevalent in adults”: this cannot be concluded from the data with certainty: the difference between adults and children was not significant and actually seemed very minor. The CI are almost completely overlapping. This should be rephrased; perhaps include other studies where they did find significant differences with age.

L600: I don’t understand the “common history” between pigs and human. 

L645-647: “water source”: it’s not so clear to which results this is referring to: it seems that river water is protective according to Table 5 (relative to stand pipe). Also, why is water source not included in the initial model in table 6 since p<0.2 according to table 5?

L675: I don’t think any results/analyses at household level are included in the paper?

Reviewer #2: Conclusions are supported by the data presented, but as Taenisasis/Cysticercosis is also a zoonose. This study rise even more question to address this public health problem. Beyond determination of prevalence in remote area, this work focus upon genetic of taenia solium and lack of perspective in order to give valuable data to fight taenia solium in this area and in low and middle income countries in general. Strains of taenia solium have been clearly identified and genotyped, but neither characteristic linked to the intermediate hosts (pork) nor the neurological conditions or pathogenicity (burden) in human participants been evaluated. Moreover, more information upon local habits and environment could help to explain the results of multivariate analyses. As an example, during deworming campaigns, taenia carriers are expected to expel scolex and proglottids, the latter need to be correctly treated in order to be sure to efficiently destroy eggs and avoid a massive contamination of environment and particularly the water used and ingested by both humans and livestock. Inefficient treatment of gravid proglottids following mass treatment can lead to a counter intuitive augmentation of cases in area. Identification of tapeworm carriers remains a challenging problem in community and anti-helminthic treatment could help in such. Authors broach the subject of self-infestation (Cysticercosis) in tapeworms carriers but diagnostic of infections are not presented for the 11 carriers identified in the study nor for the people living in the same household while a higher rate of exposure and infection are expected.

Reviewer #3: The conclusions are supported by the data presented and limitations of analysis are clearly described. Data are public health relevance addressed.

**Editorial and Data Presentation Modifications?**

Reviewer #1: Author summary: I’m not sure if this is understandable enough for the lay public

L86: associated “with”?

L92-93: the life cycle is not maintained among humans. This is confusing and should be rephrased

L139-141: this sentence is not clear to me. Perhaps explain fokontany already here (first use of the word).

L179: common instead of commonly

L194: “spp.” should not be in italics (a species name should be in italics, but the word species not)

L304: use sensitivity (not sensibility)

L341: this sentence is confusing as it reads “p > 0.05 as statistically significant”, whereas values < 0.05 are considered statistically significant. It should be rewritten

L352: diagnosis instead of diagnostic?

Table 1: why not just write “4 to 6” instead of “4 ≤ pers. ≤ 6”?

L405-409: I think this is material and methods (or introduction), but not results

Fig 3 caption: not really “from Taenia eggs of human stool”, but rather DNA extraction of stool samples in which eggs were detected? Based on M&M, it seems that DNA was extracted from the stool sample, not only the eggs? Same for L429

L479: cysticerci (spelling mistake)

L496 “behaviors factors” (grammar)

L497: Table (not tables)

L499: these are methods (not results)

L501: school attendance = education? I would suggest to use the same wording as in table 4. I don’t understand the * for education in Table 4, since the p value is > 0.2. I assume it’s a mistake?

Table 4: there seem to be many more questions compared to Table 1? Why are some of these questions also included in Table 1, but not others?

L508-509: This is confusing, since this seems to be a mixture of both EITB and Ag ELISA results? 

L538: grammar

L545: ranging between what? Between villages/districts/…?

L557-565: this is a long and complicated sentence. Please rephrase it. Also, I agree that you should only compare prevalence estimates among studies with a comparable methodology and that microscopy might result in a lower prevalence estimate, but “weak” isn’t the best word for it (if that’s what is meant in the last part of the sentence) . 

L565-566: I don’t think the comparison with Honduras is very relevant? Is it just because the point estimate almost the same, or are there other reasons to specifically compare to Honduras? Keep in mind that your 95% CI was 1.3 – 4.3, so it might also be “similar” to other studies above? I also think that this part can be merged with the next paragraph.

L614: I don’t understand the connection between “… Brazil (Figure 3).” and “The description of T. solium…”. Add a sentence to connect both parts or make a new paragraph.

L629: this sentence suggests that antibodies are lower in people with epilepsy compared to the general population. Rephrase and add more references for the seroprevalence in people with epilepsy.

L634: replace “à” by “to”; “weak” is not correct

Reviewer #2: Taeniasis/Cysticercosis by taenia solium are diseases which need a multidisciplinary approach to be fought. Authors would consider their study in a more one-health approach and would discuss additional research to be perform in this specific area with the inherent field limitations and highlight in what their result contribute to eradication.

Reviewer #3: Minor Revision

The manuscript should review by a native english speaker to improve the quality of the written.

**Summary and General Comments**

Reviewer #1: This is an important and interesting study that describes the prevalence of taeniosis and cysticercosis in a remote area in Madagascar. The design and methods are appropriate, but certain aspects should still be clarified (such as the randomisation process and (an estimate) of the total population in these villages, in order to be able to assess the representativeness of the analysis set for the total population, and more details about the questionnaire that was used). The authors should also be more nuanced when drawing conclusions from the data.

Reviewer #2: Although this study give some interesting information upon the realization of a survey upon the use of correct diagnostic test and the sharp genetic identification of parasites from a deprived setting, little is discussed on how about these assays and data could be usefully mobilized to give a dedicated response to the burden of tapeworm infection by Taenia solium in remote areas of Madagascar.

Reviewer #3: This is the first large-scale study in Madagascar to examine the association between potential risk factors measured at the individual-, household-, and village-level and the prevalence of Taenia solium taeniasis/cysticercosis in humans. The authors showed high rates of taeniasis cases affecting all ages in both townships investigated, strongly exposing human and pigs to cysticercosis. This study has a sufficient sample size, well-designed analysis and is quite robust. However, a native english speaker review should improve the quality of writing.

PLOS authors have the option to publish the peer review history of their article (what does this mean?). If published, this will include your full peer review and any attached files.

Reviewer #1: No

Reviewer #2: No

Reviewer #3: No
---

## [Decision Letter · Decision Letter 1]

23 Nov 2021

Dear Dr Rahantamalala,

Thank you very much for submitting your manuscript "Prevalence and factors associated with human Taenia solium taeniasis and cysticercosis in twelve remote villages of Ranomafana rainforest, Madagascar" for consideration at PLOS Neglected Tropical Diseases. As with all papers reviewed by the journal, your manuscript was reviewed by members of the editorial board and by several independent reviewers. The reviewers appreciated the attention to an important topic. Based on the reviews, we are likely to accept this manuscript for publication, providing that you modify the manuscript according to the review recommendations. 

Please address minor comments from Reviewer 1 and then it will be ready for acceptance. 

Sincerely,

Jessica K Fairley, MD, MPH

Associate Editor

Aaron Jex

Deputy Editor

Reviewer's Responses to Questions

**Key Review Criteria Required for Acceptance?**

**Methods**

-Are the objectives of the study clearly articulated with a clear testable hypothesis stated?

-Is the study design appropriate to address the stated objectives?

-Is the population clearly described and appropriate for the hypothesis being tested?

-Is the sample size sufficient to ensure adequate power to address the hypothesis being tested?

-Were correct statistical analysis used to support conclusions?

-Are there concerns about ethical or regulatory requirements being met?

Reviewer #1: The changes made by the authors made the methodology more clear.

Reviewer #2: The objectives are clearly stated, the study design is appropriate. The population is well described, the sample size and the statistical analyses do not call for any remarks. The ethical and regulatory procedures appear to have been correctly carried out.

**Results**

-Does the analysis presented match the analysis plan?

-Are the results clearly and completely presented?

-Are the figures (Tables, Images) of sufficient quality for clarity?

Reviewer #1: The results are generally clear, though the analysis at household level is not clear.

Table 3: I don’t fully understand “Village per Taenia carrier and per household (mHH included/Total mHH))”. There were 11 tapeworm carriers, but 41 households in this subgroup, so the subset is not clear to me. Also in L480, the sentence and the subgroup is not clear. It could also be an estimate at household level (if one person is positive, the household is positive because the sentence starts with “At household level”)? This should be clarified.

Reviewer #2: The results, figures and tables are clearly presented.

**Conclusions**

-Are the conclusions supported by the data presented?

-Are the limitations of analysis clearly described?

-Do the authors discuss how these data can be helpful to advance our understanding of the topic under study?

-Is public health relevance addressed?

Reviewer #1: (No Response)

Reviewer #2: The authors' conclusions are supported by their results. The public health relevance and perspectives of this work are clearly and appropriately addressed in this revised version of the manuscript.

**Editorial and Data Presentation Modifications?**

Reviewer #1: L390: write 46 in full; the sentence is not correct/not clear. Rephrase.

Table 3: given the extremely wide CI per village, it’s not appropriate to report decimals for the proportions, because the estimates are very uncertain.

L622-627: I still find the hypothesis of common history of people and pigs very strange. I would suggest to remove it (or clarify). 

L657 and further: “In our study villages, washing hands even with soap seems to expose participants to contaminated eggs”: you determined an association; this is not necessarily a causal relation. This should be rephrased. I assume there are likely confounding factors. 

L1163: how were these 8 active cases determined? I assume these are the 8 sero Ag people in the unclear subgroup.

English should still be improved.

Reviewer #2: in spite of some somewhat "academic" passages, which could be lightened, I recommend to accept this revised version of the manuscript.

**Summary and General Comments**

Reviewer #1: (No Response)

Reviewer #2: (No Response)

PLOS authors have the option to publish the peer review history of their article (what does this mean?). If published, this will include your full peer review and any attached files.

Reviewer #1: No

Reviewer #2: No

Figure Files:

Data Requirements:

Reproducibility:

References

---

## [Editor Report · Decision Letter 2]

17 Feb 2022

Dear Dr Rahantamalala,

We are pleased to inform you that your manuscript 'Prevalence and factors associated with human Taenia solium taeniasis and cysticercosis in twelve remote villages of Ranomafana rainforest, Madagascar' has been provisionally accepted for publication in PLOS Neglected Tropical Diseases.

Best regards,

Jessica K Fairley, MD, MPH

Associate Editor

Aaron Jex

Deputy Editor

---

## [Editor Report · Acceptance letter]

31 Mar 2022

Dear Dr Rahantamalala,

We are delighted to inform you that your manuscript, "Prevalence and factors associated with human Taenia solium taeniasis and cysticercosis in twelve remote villages of Ranomafana rainforest, Madagascar," has been formally accepted for publication in PLOS Neglected Tropical Diseases.

Best regards,

Shaden Kamhawi

co-Editor-in-Chief

Paul Brindley

co-Editor-in-Chief
